# Mapping the cardiac vascular niche in heart failure

Fabian Peisker [1,11], Maurice Halder [1,11], James Nagai [2,3], Susanne Ziegler[1], Nadine Kaesler[1,4], Konrad Hoeft[1], Ronghui Li[2,3], Eric M. J. Bindels [5], Christoph Kuppe [1], Julia Moellmann[6], Michael Lehrke[6], Christian Stoppe[7], Michael T. Schaub [8], Rebekka K. Schneider [9], Ivan Costa [2,3,12] & Rafael Kramann [1,9,10,12✉]

The cardiac vascular and perivascular niche are of major importance in homeostasis and during disease, but we lack a complete understanding of its cellular heterogeneity and alteration in response to injury as a major driver of heart failure. Using combined genetic fate tracing with confocal imaging and single-cell RNA sequencing of this niche in homeostasis and during heart failure, we unravel cell type specific transcriptomic changes in fibroblast, endothelial, pericyte and vascular smooth muscle cell subtypes. We characterize a specific fibroblast subpopulation that exists during homeostasis, acquires *Thbs4* expression and expands after injury driving cardiac fibrosis, and identify the transcription factor TEAD1 as a regulator of fibroblast activation. Endothelial cells display a proliferative response after injury, which is not sustained in later remodeling, together with transcriptional changes related to hypoxia, angiogenesis, and migration. Collectively, our data provides an extensive resource of transcriptomic changes in the vascular niche in hypertrophic cardiac remodeling.

[1] Institute of Experimental Medicine and Systems Biology, RWTH Aachen University Medical Faculty, Aachen, Germany. [2] Institute for Computational Genomics, RWTH Aachen University Hospital, Aachen, Germany. [3] Joint Research Center for Computational Biomedicine, RWTH Aachen University Hospital, Aachen, Germany. [4] Division of Nephrology and Clinical Immunology, RWTH Aachen University Medical Faculty, Aachen, Germany. [5] Department of Hematology, Erasmus MC Cancer Institute, Rotterdam, The Netherlands. [6] Department of Cardiology, RWTH Aachen University Hospital, Aachen, Germany. [7] Department of Anesthesiology, Intensive Care, Emergency, and Pain Medicine, University Hospital Wuerzburg, Wuerzburg, Germany. [8] Department of Computer Science, RWTH Aachen University, Aachen, Germany. [9] Institute of Cell Biology, RWTH Aachen University Hospital, Aachen, Germany. [10] Department of Internal Medicine, Nephrology and Transplantation, Erasmus Medical Center, Rotterdam, The Netherlands. [11] These authors contributed equally: Fabian Peisker, Maurice Halder. [12] These authors jointly supervised this work: Ivan Costa, Rafael Kramann. ✉email: rkramann@gmx.net

Heart failure (HF) is a leading cause of morbidity and mortality worldwide[1]. Coronary microvascular dysfunction is an important mechanism in cardiac pathophysiology during HF, where diffuse capillary loss and adverse remodeling of intramural cardiac arterioles throughout the left ventricular myocardium triggers hypoxia and fibrosis[2]. The vascular and perivascular niche of the mammalian heart harbors, in addition to the endothelium, important mesenchymal cell types including mural cells (vascular smooth muscle cells (VSMC) and pericytes) as well as fibroblasts[3]. The localization and heterogeneity of these cell types throughout the vascular network of the heart—from large arteries and arterioles to small capillaries and then toward the venous system—is only partially known. In particular, detailed insights into how the perivascular niche changes during cardiac remodeling in HF is lacking. Fibrosis is present in almost every form of heart disease and several studies indicate a perivascular origin of fibrosis driving myofibroblasts[4]. Prior to fibrosis, angiogenesis is an early attempt of the injured heart to compensate for increased oxygen consumption of hypertrophic cardiomyocytes which later transitions into capillary loss, cell death and replacement fibrosis[5]. Mural cells are of key importance in the vascular niche as they provide stability and elasticity to blood vessels and regulate flow by their contractile properties with an important role for myocardial perfusion in homeostasis in congestive HF[6]. Various studies have demonstrated plasticity of stromal and endothelial cells particularly in disease[7–11]. However, due to the strong heterogeneity of the vascular niche, the individual contributions of different vascular and perivascular cell types to cardiac remodeling remain unclear.

The goal of this study is to dissect the heterogeneity and individual contribution of stromal and endothelial cell types to cardiac remodeling in HF by combining inducible genetic fate tracing with single-cell RNA sequencing and confocal imaging. For this we use various mouse lines for genetic fate tracing of cardiac endothelial, pericyte, VSMC and fibroblast lineages. This approach enables us to capture and map cell type specific transcriptomic changes over two independent time points in hypertrophic left ventricular remodeling. Our data indicate that all cardiac fibroblast populations shift toward an extracellular matrix (ECM) expressing profile after injury. However, one specific fibroblast subtype marked by *Thbs4* expression shows significant expansion, likely differentiated from a broader pool of fibroblasts and expresses distinct ECM signature. Endothelial cells respond after injury by increased proliferation, but fail to sustain angiogenesis in later remodeling. Taken together, we provide an extensive overview of cell type specific transcriptomic changes in the cardiac vascular niche in hypertrophic remodeling.

## Results

**Fate tracing and scRNA-seq mapping of the vascular niche in early heart failure.** The vascular and perivascular niche of the heart harbors various cell types that are critical for the cardiac function in homeostasis and disease. However, their heterogeneity, crosstalk and fate during HF remains partly unclear. Here, we combined inducible genetic lineage tracing with high-resolution confocal imaging and scRNA-seq, to trace vascular and perivascular cell types of the heart in murine pressure overload induced HF. Various transgenic *CreER;Rosa26tdTomato* (*tdTom*) mouse lines that recombine in different cell populations of the vasculature and perivasculature were pulsed with tamoxifen to induce recombination (fibroblasts: Collagen Type I Alpha 1 Chain—*Col1a1CreER*; fibroblasts and mural cells: Platelet Derived Growth Factor Receptor Beta—*PdgfrβCreER*; Gli1[+] progenitors: Glioma-Associated Oncogene 1, *Gli1CreER*; endothelium: Cadherin 5, *Cdh5CreER*; VSMC and pericytes: Myosin

Heavy Chain 11—*Myh11CreER* and Chondroitin Sulfate Proteoglycan 4 ((*Cspg4*), *Ng2CreER*). After a 21 day washout period, the mice were subjected to either sham or transverse aortic constriction (TAC) surgery (Fig. 1a, b). Mice were killed either 2 (all lines) or 4 weeks (*Gli1CreER;tdTomato* and *Cdh5CreER;td-Tomato*) after surgery. We observed increased heart weight (Fig. 1c), interstitial fibrosis (Fig. 1d) and functional decline measured by echocardiography (in *Gli1CreER;tdTomato* and *Cdh5CreER;tdTomato* lines, Supplementary Fig. 1a) as hallmarks of cardiac remodeling in this model. We used the apical half of the left ventricle (including apex, anterior-, lateral-, posterior-wall and septum) to generate a single-cell suspension with subsequent FACS enrichment for tdTom[+], viable (DAPI[−]) cells for scRNA-seq (10x Genomics). The basal part of the left ventricle was used for confocal imaging analysis (Fig. 1e).

The fate traced cells showed partially distinct and partially overlapping localizations (Fig. 1f and Supplementary Fig. 1b–h). In homeostasis tdTom[+] cells from *Col1a1*, *Gli1* and *Pdgfrβ* lineage were located in the perivascular and adventital region of arteries or in the cardiac interstitium adjacent to endothelial cells of myocardial capillaries. While *PdgfrβCreER* also recombined in the VSMC layer only few VSMC were labeled in the *Gli1CreER* mice. *Cdh5CreER* showed strong and specific recombination of all cardiac endothelial cells including endothelium of large vessels and capillaries. Both *Myh11CreER* and *Ng2CreER* labeled cells within the VSMC layer of larger vessels and further interstitial cells adjacent to myocardial capillary endothelium as expected. We did not observe strong localization changes of fate traced cells 2 weeks after TAC surgery. However, we noticed focal expansion of cells from *Col1a1*, *Gli1* and *Pdgfrβ* lineage following injury, with *Pdgfrβ* lineage derived cells showing the clearest expansion, in line with an early interstitial fibrosis (Fig. 1f and Supplementary Fig. 1g, h).

We generated in total 14 individual single-cell libraries from sorted tdTom[+] cells of each transgenic line and pooled 3–5 mice for each library. All transgenic lines were used to generate a library from either sham or TAC after 14 days and in addition we generated libraries from *Gli1CreER;tdTomato* and *Cdh5CreER;td-Tomato* mice 28 days after TAC (Fig. 1b–e). After filtering the individual datasets for cell quality, we integrated all libraries using batch correction by Harmony[12] (Supplementary Fig. 2a). Harmony outperformed other integration methods in multiple recent benchmarking studies[13,14], and performed well for our dataset. Filtering for *tdTom* mRNA expression ensured solely inclusion of genetically traced cells (Supplementary Figs. 2b and 3a). The final dataset of the full integration included 77602 high quality *tdTom* expressing cells and unbiased clustering detected seven distinct major clusters (Fig. 2a). We clearly identified the three major cell types of the cardiac vascular niche: fibroblasts (*Dcn*[+], *Col1a1*[+], *Pdgfrα*[+]), endothelial cells (*Kdr*[+], *Pecam1*[+]), and mural cells (*Rgs5*[+], *Kcnj8*[+], *Vtn*[+]). Mural cells included pericytes (*Abcc9*[+], *Colec11*[+]) and VSMC (*Acta2*[+], *Tagln*[+]) (Fig. 2b and Supplementary Fig. 3b). We additionally detected a few small clusters identified as Schwann cells (*Kcna1*[+], *Plp1*[+]), lymphatic endothelial cells (*Ccl21a*[+], *Mmrn1*[+]), proliferating endothelial cells (*Top2a*[+], *Pecam1*[+]) (Supplementary Fig. 3c, d) and one distinct cluster characterized by a strong interferon response signature (*Itif1*[+], *Itif3*[+]) (Fig. 2a, b). Interestingly, this signature led to joint clustering of otherwise distinct cell types, including fibroblasts, endothelial cells and pericytes suggesting that the interferon response genes influenced the clustering stronger as compared to cell identity. Moreover, subclusters of fibroblasts and endothelial cells defined by interferon response genes have been described in other scRNA-seq studies of the murine heart[15–17].

We next compared the contribution of all genetically fate traced cell populations to each major cell type identified in the

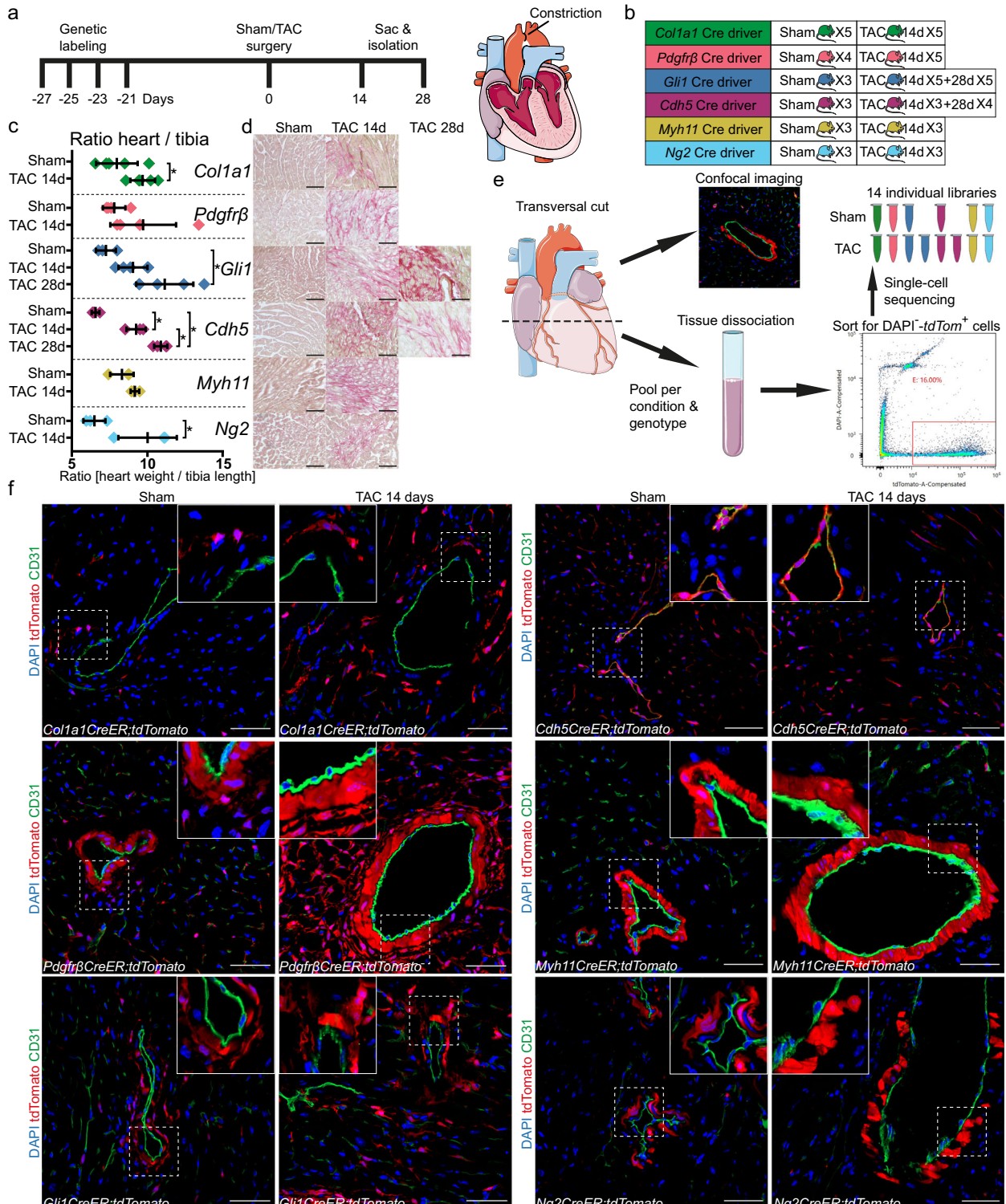

**Fig. 1 Fate tracing of the vascular niche after TAC induced injury. a** Experimental timeline of genetic labeling of specific cell types using tamoxifen, 3-week washout period, surgery time point (transverse aortic constriction (TAC) or sham) and duration until harvest. The TAC procedure is illustrated. **b** Overview of genetic Cre driver mouse lines and animal numbers per group. **c** Heart weight to tibia length ratio between TAC and sham groups. Each dot represents an individual mouse (mean ± SD; number of independent mice per group is displayed in **b**; *$p < 0.05$; unpaired *t*-test, two-sided). Source data are provided as a Source Data file. **d** Representative pictures of picro-sirius red staining for TAC (14 and 28 days) and sham hearts, one per genotype (scale bar 50 μm). **e** After extraction murine hearts were cut transversely to separate the upper part for imaging and lower part from basal region to apical area for dissociation. Prior to scRNA-seq samples were pooled per group, time point and genotype. Single-cell suspensions were DAPI stained, followed by FACS sorting for DAPI⁻ and tdTom⁺ cells. Fourteen individual single-cell cDNA libraries were generated using the 10x genomic platform. **f** Representative stainings visualizing localization of fate traced cell types (tdTomato, red) in TAC and sham (scale bars 50 μm). The Figure was partly generated using Servier Medical Art, provided by Servier, licensed under a Creative Commons Attribution 3.0 unported license.

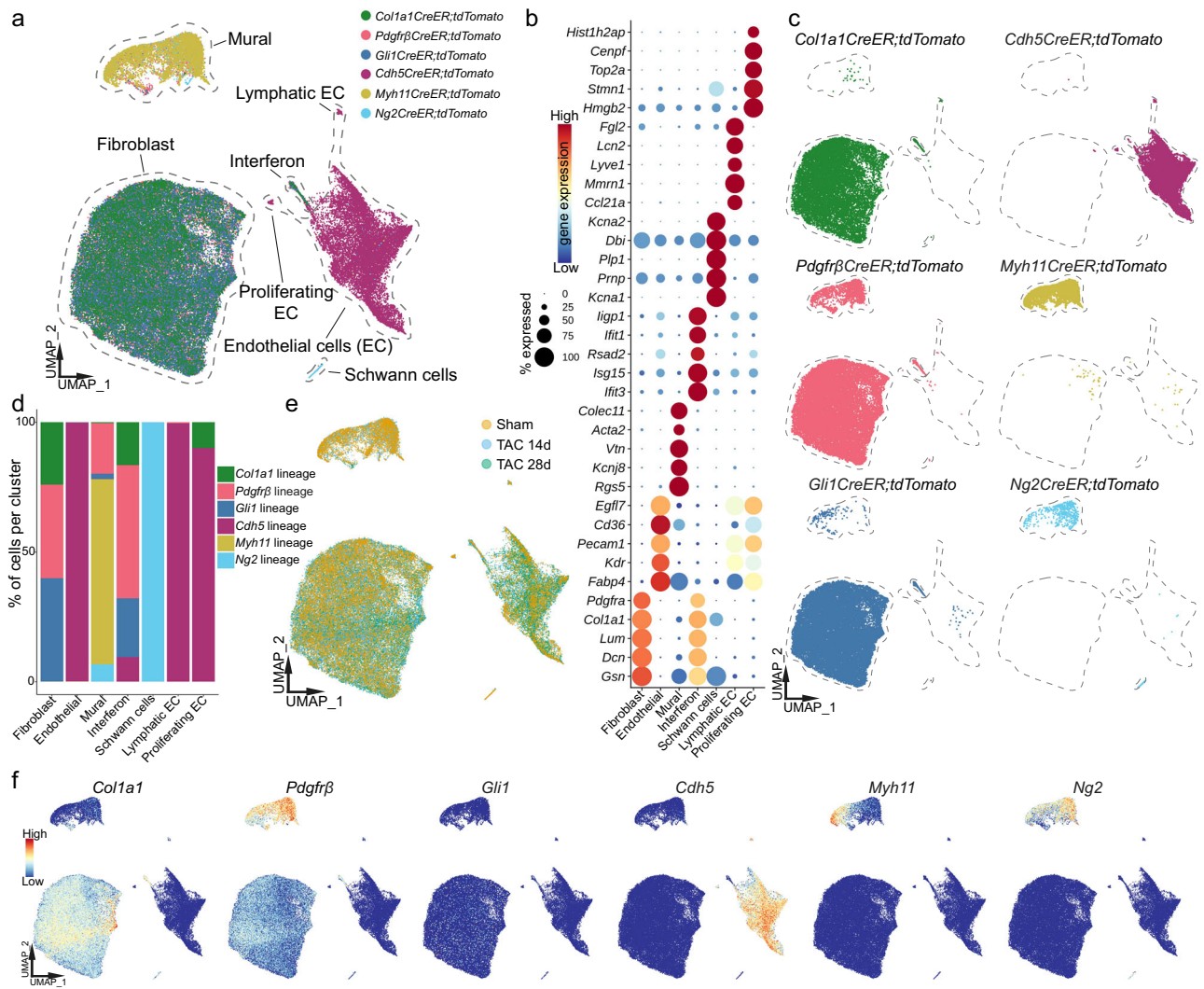

**Fig. 2 Single-cell sequencing of fate traced vascular niche cells. a** UMAP representation of 77,602 cells based on the integration of all 14 individual scRNA-seq libraries. Dot colors indicate the original fate traced genotype of each cell. Major cell type annotations are included. **b** Dot plot of top five expressed marker genes per cluster. Dot size refers to proportion of cells expressing the gene per cluster. **c** UMAP embedding as in **a**, split up according to lineage tracing genotype (sham and TAC conditions combined per genotype). **d** Bar graph showing contribution of the different lineage tracing genotypes to the major cell type clusters. **e** UMAP embedding labeling cells according to underlying condition (sham/TAC). **f** UMAP plots of selected genes used in the induced lineage tracing models, depicting their expression pattern. Scaled gene expression is indicated by color.

scRNA-seq analysis (Fig. 2c, d). The fibroblast cluster primarily consisted of cells originating from the *Col1a1*, *Gli1* or *Pdgfrβ* lineage, while endothelial cells were primarily *Cdh5* lineage derived and mural cells from *Myh11*, *Ng2*, *Gli1* and *Pdgfrβ* lineage (Fig. 2c, d). Schwann cells were solely *Ng2* lineage derived (Fig. 2c, d). Interestingly, the fibroblasts that were derived from different fate traced lineages were all evenly distributed within the fibroblast cluster, suggesting similarity despite using different markers for fate tracing. We did not observe any major cell type to be exclusive or have a bias for either sham or TAC (Fig. 2e). In addition, we observed higher *Pdgfrβ* mRNA expression within the mural cell cluster (Figs. 1f and 2f).

We would like to point out that the *Pdgfrβ* lineage derived cells considerably expanded around larger myocardial vessels following TAC, while we did not observe such expansion of *Myh11* or *Ng2* fate traced cells in this location (Fig. 1f and Supplementary Fig. 1f). This suggests that the expanding cells in TAC are primarily of fibroblast and not of mural cell origin. Most genetically fate traced cells still showed expression of their marker gene used for the Cre recombination (Fig. 2f). Only for *Myh11*,

we primarily observed mRNA expression in VSMC while the *Myh11CreER* fate traced cells contributed to both VSMC and pericytes (Fig. 2c, d, f and Supplementary Fig. 3b). This can be explained by the fact that scRNA-seq data is sparse and biased to detection of highly expressed genes, while the *Myh11* Cre driver is strong and also recombines in cells with a low *Myh11* gene expression. Additionally, *Myh11CreER* lineage tracing was recently shown to label pericytes even in the smallest capillaries of the retina[18]. Taken together, we were able to map cells from all major vascular and perivascular cell types of the murine heart in homeostasis and pressure overload induced HF at unprecedented resolution.

**Heterogenous fibroblast populations show plasticity in cardiac remodeling and drive fibrosis.** Cardiac fibrosis is the process of ECM deposition within the heart and occurs after virtually any injury to the heart. Fibrosis leads to cardiac dysfunction by increasing stiffness and also triggering contractile dysfunction and electric instability[4]. In order to identify and characterize the

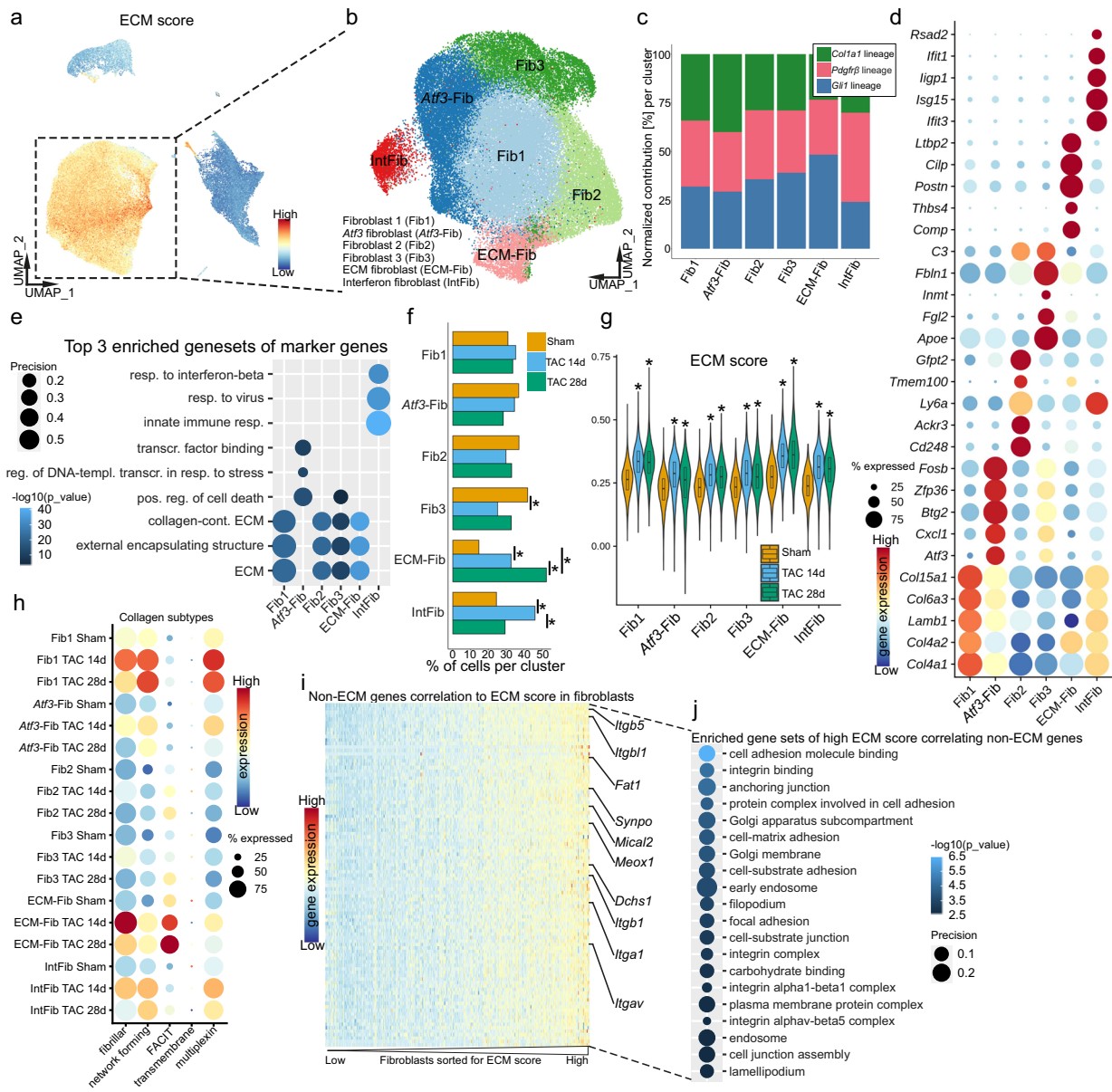

**Fig. 3 Subclustering of fibroblasts and subsequent analysis of ECM expression. a** UMAP embedding showing extracellular matrix (ECM) score in the overall integrated dataset. **b** UMAP of subsequent subclustering of 54905 fibroblasts, consisting of *PdgfrβCreER, Gli1CreER, Col1a1CreER* derived cells. Cluster annotation: fibroblast 1 (Fib1), *Atf3* fibroblasts (*Atf3*-Fib), fibroblast 2 (Fib2), fibroblast 3 (Fib3), ECM fibroblast (ECM-Fib), interferon fibroblast (IntFib). **c** Normalized contribution of each fate traced genotype to the corresponding cluster. **d** Dot plot of top five expressed marker genes per cluster. Dot size refers to proportion of cells expressing the gene per cluster. **e** Top three representative gene sets enriched (hypergeometric test, one-sided) in marker genes per fibroblast subtype. Dot size refers to overlap of tested genes and gene set (precision). Full list in Supplementary Data 1. **f** Normalized proportion of fibroblasts per subcluster from sham and TAC (*: false discovery rate (FDR) <0.05 and absolute log2 fold change >0.58, see also Supplementary Fig. 4e). **g** Violin plot of ECM scores per condition and fibroblast cluster (all fibroblast Cre drivers combined). Integrated boxplots show center line as median, box limits as upper and lower quartiles. All clusters show significant difference between TAC and sham, ECM-Fib show significant differences to all other subtypes (*p < 0.001, two-sided Wilcoxon rank sum test, unpaired, Bonferroni adjusted p value). **h** Dot plot of expression scores for collagen subtypes per fibroblast cluster separated by condition and time point. Dot size refers to proportion of cells expressing the gene set per cluster. **i** Heatmap of top 100 non-ECM related genes correlating in gene expression to the ECM score of all fibroblasts. Fibroblasts are sorted by ECM score on the *x*-axis from low-to-high score. Genes are listed on the *y*-axis, color indicates gene expression level. **j** Top 20 genes sets enriched (hypergeometric test, one-sided) in the top 100 genes correlating to ECM score from **i**. Dot size refers to overlap of test genes and gene set (precision). For details on statistics and reproducibility, see "Methods".

major cellular source of myocardial ECM after injury, we utilized an ECM score that contains mRNA expression of all known ECM collagens, glycoproteins and proteoglycans[19,20]. This approach clearly identified fibroblasts as the cell type with the highest ECM gene expression in our mouse model, in particular with regards to

collagens and proteoglycans (Fig. 3a and Supplementary Fig. 3e). We thus focused first on the fibroblasts and clustered these cells separately to increase resolution and gain a deeper understanding of the heterogeneity and regulation of matrix producing cells in cardiac remodeling. This led to the identification of six

subclusters with a total of 54,905 fibroblasts, which showed considerable differences in gene expression suggesting potentially biological meaningful heterogeneity (Fig. 3b and Supplementary Fig. 2a, c). We defined the following fibroblast subtypes for further analysis: fibroblast 1 (Fib1; higher in collagens, e.g., Col4a1/2⁺), Atf3 fibroblasts (Atf3-Fib; Atf3⁺, Cxcl1⁺, Mt1⁺), fibroblast 2 (Fib2; CD248⁺, Ackr3⁺), fibroblast 3 (Fib3; Fgl2⁺, Inmt⁺), ECM fibroblast (ECM-Fib; Postn⁺, Comp⁺) and interferon fibroblast (IntFib; Ifit1⁺, Ifit3⁺) (Fig. 3b–d). The three integrated lineages (Gli1, Pdgfrβ and Col1a1) contributed to all subsets almost equally with two exceptions. The Pdgfrβ lineage contributed more to the IntFib, whereas the Gli1 lineage contributed more to the ECM-Fib (Fig. 3c and Supplementary Fig. 4a). Gene set enrichment analysis (GSEA) based on the subtype marker genes (Fig. 3d) suggested shared basic functions for all fibroblast subtypes, whereas Atf3-Fib marker genes were particularly associated with stress response and apoptosis, IntFib with interferon response and ECM-Fib with a more pronounced ECM production (Fig. 3e and Supplementary Data 1).

To verify these data with an orthogonal method and dissect if there are specific localizations of these identified fibroblast subtypes, we performed in situ hybridization of specific marker genes on left ventricular tissue. Separate co-stainings for Ifit1, Fgl2 and Atf3 with tdTom, confirmed the co-expression of these marker genes in Gli1 or Pdgfrβ lineage derived cardiac fibroblasts (Supplementary Fig 4b). For Ifit1, we observed tdTom co-staining in focal interstitial areas (Supplementary Fig. 4b). Fgl2 and Atf3 were expressed in fibroblasts throughout the left ventricular tissue with no noticeable enrichment in areas of fibrosis or larger vessels (Supplementary Fig 4b). Atf3 has already been described as a marker gene for cellular stress response induced by different stimuli[21]. More specifically within the heart, Atf3 was reported to be elevated toward the terminal stage of the mitochondrial integrated stress response, suggesting the Atf3-Fib subtype might be involved in this response[22].

We next asked whether fibroblast subtypes differ in their abundance in cardiac homeostasis and injury (Fig. 3f and Supplementary Fig. 4c, d). Interestingly, ECM-Fib continuously increased significantly after TAC, while IntFib initially increased after 14 days and then decreased again at 28 days. For the remaining subtypes we did not observe significant changes in our composition analysis, except for a decrease of Fib3 after 14 days (Fig. 3f and Supplementary Fig. 4d). Analysis of cell cycle associated gene expression yielded no detection of fibroblast proliferation, suggesting that ECM-Fib might expand through differentiation rather than proliferation (Supplementary Fig. 4e). A similar finding was reported by McLellan and colleagues in a murine model of Angiotensin 2 induced cardiac remodeling (AngII model)[23].

To further analyze whether fibrotic ECM deposition is primarily driven by specific fibroblast subtypes, we compared ECM expression across fibroblast subtypes. The ECM score was largely similar in all fibroblast subpopulations in homeostasis, while after TAC specifically the ECM-Fib cluster showed the highest ECM score, followed by Fib1 and IntFib (Fig. 3g and Supplementary Fig. 4f). Comparing particular collagens including different functional groups of collagens[24], revealed Fib1 and ECM-Fib as highest collagen expressing fibroblasts (Fig. 3h and Supplementary Fig. 4g, h). ECM-Fib showed strong expression of fibrillar collagens (e.g., Col1a1/2) 14 days after TAC, which declined at 28 days. Fib1 on the other hand primarily expressed network forming (e.g., Col4/6 subtypes) and multiplexin collagens (e.g., Col15a1) after injury.

By correlating gene expression values to the ECM score in fibroblasts, we identified genes and cellular processes that are potentially involved in the regulation of ECM expression (Fig. 3i).

Interestingly, we found Meox1 among the 100 highest correlating genes, which was recently published as an important regulator of fibroblast activation[25,26]. In addition, several integrin subtypes showed a high correlation and GSEA based on correlating genes detected multiple cell adhesion related GO terms (Fig. 3j). Several studies have demonstrated the importance of integrins as mediators of organ fibrosis and αv-integrin as a potential target to prevent fibrosis progression[27,28]. In Summary, we were able to define fibroblast heterogeneity in our dataset, detect subtype specific changes in tissue abundance and ECM expression during cardiac remodeling.

**Differential gene expression analysis reveals Thbs4 as marker of injury-related ECM-Fib.** To further explore heterogeneity of fibroblast subtypes during cardiac remodeling, we explored differences in gene expression (using MAST[29]), comparing each subtype between sham, TAC 14 days and TAC 28 days. After clustering significantly upregulated genes, we performed GSEA based on the identified distinct upregulated expression clusters (Fig. 4a). The differential gene expression analysis indicated partly overlapping responses of fibroblast subtypes to injury (Fig. 4a). Genes in cluster 1 were upregulated 14 days after TAC and functionally associated with collagen remodeling and PDGFRβ signaling, whereas genes in cluster 5 and 7 were upregulated after 28 days and associated with response to external stimuli, cell-cell junction and actin binding for all fibroblast subtypes. Among the genes found to be upregulated by multiple subtypes, we found Postn, Pdgfrβ, Tgfβ1, Thbs1 and several collagens among others (Fig. 4a), that have been described previously to be involved in fibrotic cardiac remodeling[30]. Overall, these genes likely reflect a more general transcriptomic response across fibroblast populations. Interestingly, we observed specific patterns of unique differentially expressed genes (DEG) for ECM-Fib (cluster 2), Fib1 (cluster 3), Fib2 (cluster 4) and IntFib (cluster 6). These gene clusters were mostly associated with different ECM related gene sets for Fib1, Fib2 and ECM-Fib, while IntFib showed specific upregulation interferon-beta response related genes (Fig. 4a).

Since ECM-Fib represented the most injury-related fibroblast population with an increasing transcriptional response after TAC 14 and 28 days, we analyze the DEG of this subtype for highly specific genes (Fig. 4b and Supplementary Fig. 5a). Thbs4, Lox, Ddah1 and Cthrc1 were among the top DEG for this cluster and had been reported as specific for injury-related fibroblasts before[23,30,31]. Particularly, Thbs4 was highly specific for activated ECM-Fib after injury (Fig. 4c). We validated this finding by immunostaining and observed that THBS4 indeed specifically marked a subpopulation of tdTom⁺ PdgfrβCreER cells (Fig. 4d). Moreover, THBS4 expression was strict to cardiac injury, while absent in non-injured hearts (Fig. 4d). THBS4⁺-tdTom⁺ cells showed focal expansion in line with development of patchy interstitial cardiac fibrosis following TAC. This finding was similarly observed in Gli1CreER and Col1a1CreER mice (Supplementary Fig. 5b). Involvement of THBS4 in human fibrotic hearts has been reported and was recently described as a marker of activated and injury-related fibroblast by two single-cell scRNA-seq studies of human heart disease[32,33]. We validated the presence of THBS4 expression in PDGFRα⁺ fibroblasts in a tissue sample from a human HF patient by in situ hybridization (Supplementary Fig. 5c).

**Acute and chronic injury-related differences in cardiac fibroblast subtypes.** We next compared our fibroblast subtypes to subtypes defined in the recently published datasets from refs. [15,23] by correlation (Supplementary Fig. 6a, b). We were able to identify three conserved subtype clusters existing in all three

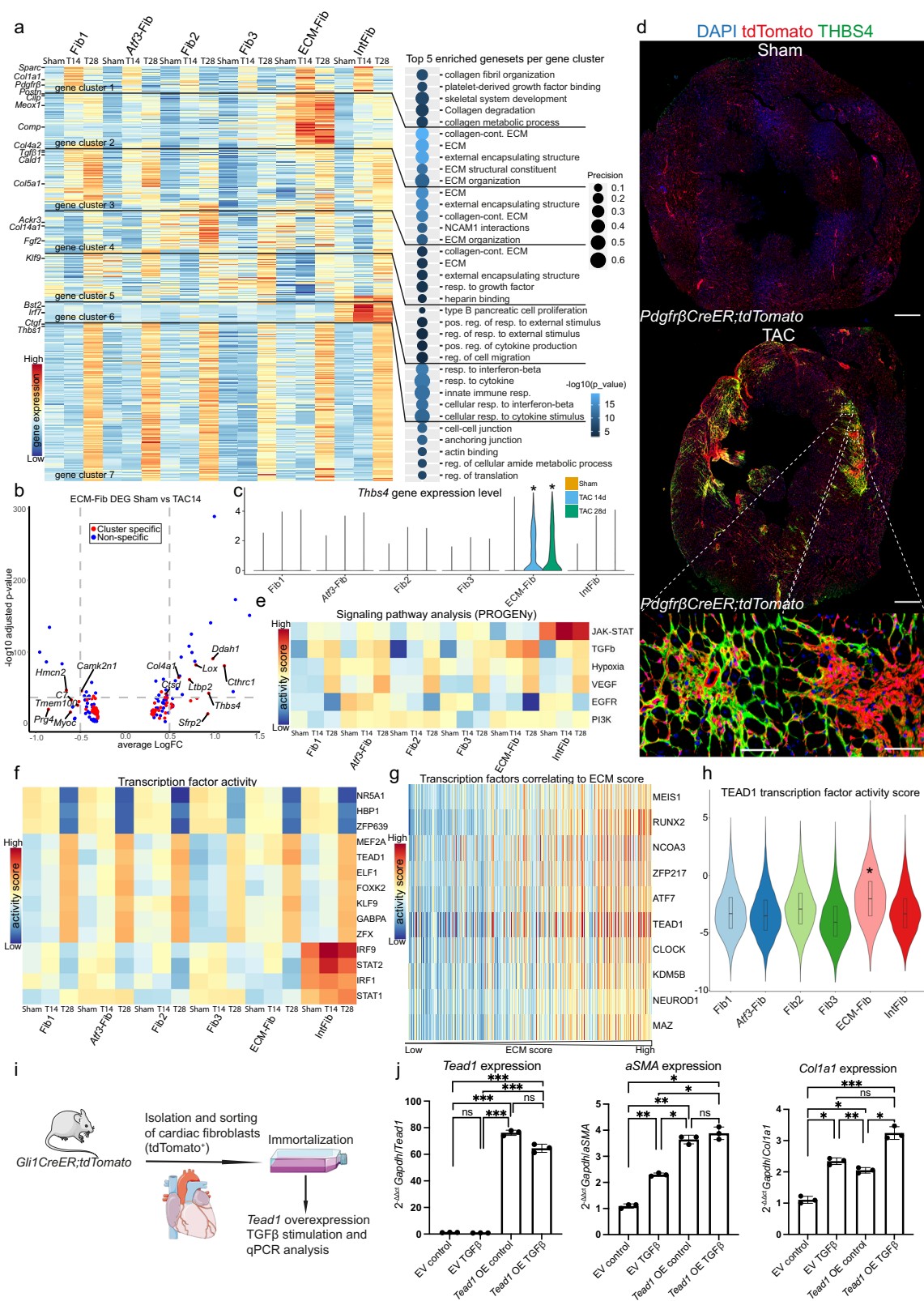

datasets: (1) fibroblasts of high ECM expression: ECM-Fib, FibThbs4/FibCilp, matrifibrocytes (MFCs); (2) interferon responsive fibroblasts: IntFib, Fib8, interferon response fibroblast (IFNr); (3) *Ly6a*[+]/*CD248*[+] fibroblasts: Fib2, Fib5, progenitor-like state fibroblast. Integration of the datasets validated this finding as correlating subtypes clustered together (Supplementary Fig. 6c, d). At this point it became apparent that one fibroblast subtype is

missing from our dataset, due to dissimilarity in study design. *Wif1*[+]/*Dkk3*[+] fibroblast seem to be located specifically in the region of heart valves, the part of the heart which in our study was only used for imaging, while Forte et al. and McLellan et al. used whole hearts (Supplementary Fig. 6e–h). Heart valve associated fibroblasts, marked by *Wif1*/*Dkk3* expression, have also been suggested as a separate subtype by refs. [3,15].

**Fig. 4 Mapping transcriptomic changes in fibroblast subtypes of highest ECM producing fibroblast, ECM-Fib. a** Hierarchical clustered gene expression heatmap of differentially expressed genes (DEG) analyzed per cluster and condition, combined with gene set enrichment analysis. DEG were identified using MAST (only upregulated genes with adjusted $p$ value <0.01 and logFC > 0.3). Top five enriched gene sets (hypergeometric test, one-sided) per gene cluster on the right. Dot size refers to overlap of tested genes and gene set (precision). **b** DEG of ECM-Fib comparing sham and TAC 14 days visualized in a volcano plot, displaying cluster specific (red) and non-specific (blue) DEG. **c** Violin plot comparing *Thbs4* expression across all fibroblast subtypes per condition (*: adjusted $p$ value <0.01, differential gene expression analysis by MAST). **d** Confocal XY scans of entire cross sections of *PdgfrβCreER;tdTomato* hearts stained for THBS4 (scale bars 500 μm, inserts 50 μm). **e** Signal pathway activity prediction of fibroblast subtypes based on pathway responsive genes (PROGENy). Color indicates relative predicted activity per pathway. **f** Transcription factor (TF) activity prediction based on TF regulons (DoRothEA) for fibroblast subtypes. Color indicates relative predicted activity per TF. **g** Correlation of TF activity scores to ECM score. Top 10 highest correlating TF displayed. Color indicates relative predicted activity per TF. Fibroblasts sorted by ECM score from low to high on the x-axis. **h** Violin plot of predicted TEAD1 transcription factor activity across fibroblast subtypes (all conditions combined). Integrated boxplots show center line as median, box limits as upper and lower quartiles. ECM-Fib shows significant differences to all other subtypes (*$p < 0.001$, two-sided Wilcoxon rank sum test, unpaired, Bonferroni adjusted $p$ value). **i** Schematic overview of cardiac fibroblast cell line generation. **j** Bar graphs of relative gene expression of *Tead1*, *Acta2* and *Col1a1* measured by RT-qPCR. Data points represent normalized expression by $2^{-\Delta\Delta Ct}$ method (mean ± SD, $n = 3$ independent experiments per group, *$p$ value < 0.05; **$p$ value < 0.01; ***$p$ value < 0.001; ns not significant; one-way ANOVA with Tukey's post hoc). Source data are provided as a Source Data file. For details on statistics and reproducibility, see "Methods". The Figure was partly generated using Servier Medical Art, provided by Servier, licensed under a Creative Commons Attribution 3.0 unported license.

One other important observation of our integration approach was the different occurrence of *Thbs4*+ high ECM expressing fibroblasts and *Acta2*+ myofibroblasts, with the latter clearly found early after myocardial infarction (MI)[15], though not in the TAC (our data) nor AngII model[23] (Supplementary Fig. 6i–l). This suggests a strong diversity in injury dependent activation of fibroblasts. Myofibroblasts seem to be required mainly in acute ischemic injury with subsequent replacement fibrosis, since contractile myofibers might be needed to contract the large wounds after MI while they might not be required in interstitial fibrosis caused by increased afterload.

**Signal pathway and TF activity prediction identifies TEAD1 as pro-fibrotic regulator.** Signaling pathway analysis indicates high JAK-STAT signaling in the IntFib cluster in sham that further increased in TAC (Fig. 4e and Supplementary Fig. 5d). Moreover, following TAC we observed increased hypoxia and VEGF signaling, as well as TGFβ signaling across fibroblast-types with the strongest activity in the ECM-Fib population (Fig. 4e).

TF activity estimation suggests stronger changes at 28 days after TAC as compared to day 14 or sham (Fig. 4f). This also corresponds to the larger number of DEG at the later time point (gene cluster 7, Fig. 4a). IntFib were estimated to have high activity of IRF1/9 and STAT1/2, in line with the interferon responsive gene expression profile and JAK-STAT activity of this subtype[34]. MEF2A and TEAD1, were among the TF with the highest change in activity comparing sham and TAC 28 days time point. MEF2A is known as a key regulator of myogenesis[35] and a mediator in cardiac remodeling[36]. We further observed increased TEAD1 activity (Fig. 4f), which is an important cofactor downstream of the hippo pathway[37]. Several studies investigated the role of hippo pathway activity in cardiac disease[38–40], with one recent study specifically showing its importance for pro-fibrotic fibroblasts activation[41]. We next correlated TF activity with the ECM score to identify TFs that might regulate ECM expression (Fig. 4g). This analysis pointed toward TF that have been previously described in fibrosis such as MEIS1[42], RUNX2[43] and NCOA3[44]. We again found TEAD1 highly associated with ECM expression. A focused visualization of the TF activity score per fibroblast subtype, clearly indicated a higher predicted TEAD1 activity in ECM-Fib and two interesting genes involved in TEAD1 activity (*Mical2*[45], *Palld*[46]), were found to be specifically expressed and upregulated in ECM-Fib (Fig. 4h and Supplementary Fig. 5e, f).

To validate a potential role of TEAD1 in fibroblasts we next generated a murine cardiac Gli1+ fibroblast cell line by sorting (FACS) of tdTom+ cells from *Gli1CreER;tdTomato* mice 2 weeks

after tamoxifen pulse with subsequent SVLargeT immortalization (Fig. 4i). Retroviral overexpression of *Tead1* induced a strong expression of *Acta2*, *Col1a1* and *Fn1*, independent of TGFβ stimulation indicating a direct effect of TEAD1 on pro-fibrotic differentiation (Fig. 4j and Supplementary Fig. 5g).

**Matrix producing fibroblasts originate from a large fibroblast pool.** We identified ECM-Fib as a fibroblast subtype with the highest matrix related gene expression and strongest expansion after TAC (Fig. 3f). Interestingly, our data suggested that a small fraction of these cells were already present in homeostasis prior to injury (sham) (Fig. 3f and Supplementary Fig. 4d). To exclude that this finding was an integration artifact, we separately clustered fibroblasts from only our sham samples as well as fibroblasts from the control data from refs. [15,23] (Supplementary Fig. 7a). Transferring the cluster labels from the complete annotation and comparison to the expression pattern of *Postn* clearly shows a population of fibroblasts in the control datasets, which resembles ECM-Fib (Supplementary Fig. 7a). In addition, in situ hybridization experiments identified *Postn* expressing fibroblasts (*Pdgfrα*+) in sham hearts (Supplementary Fig. 7b). This provides strong evidence of a, albeit minor, non-activated ECM-Fib population present in homeostasis that lacks *Thbs4* expression.

As described earlier, we did not identify proliferating fibroblasts 14 or 28 days after TAC, suggesting that differentiation could be the major process causing the ECM-Fib expansion after injury (Supplementary Fig. 4e). However, we cannot exclude that proliferation of pre-existing ECM-Fib occurs outside of the time-points covered by our scRNA-seq analysis. Analysis of RNA velocity in our fibroblast data separated by time point indicated the Fib1 cluster as a potential origin of ECM-Fib (Fig. 5a and Supplementary Fig. 7c). This was further highlighted by the calculated latent time for each condition, where transcriptional changes are first prominent in Fib1 at the 14 day time point and later (TAC 28) in ECM-Fib (Fig. 5b and Supplementary Fig. 7d). High latent time values also occur in *Atf3*-Fib, Fib2 and later in Fib3. Fib2 likely represents a fibroblast subtype, which has been described as a potential fibroblast progenitor (*Ly6a*+/*CD248*+)[15,47,48]. However, our RNA velocity analysis did not indicate a transcriptional shift from this cluster to any other. Therefore, we focused on a potential differentiation path from Fib1 to ECM-Fib using monocle 3 to calculate trajectories and select a path between these two clusters (Fig. 5c and Supplementary Fig. 7e). Buechler et al. recently reported a fibroblast population marked by Col1a15 as a cross-organ intermediate fibroblast subtype, between a potential Pi16+ progenitor population and different

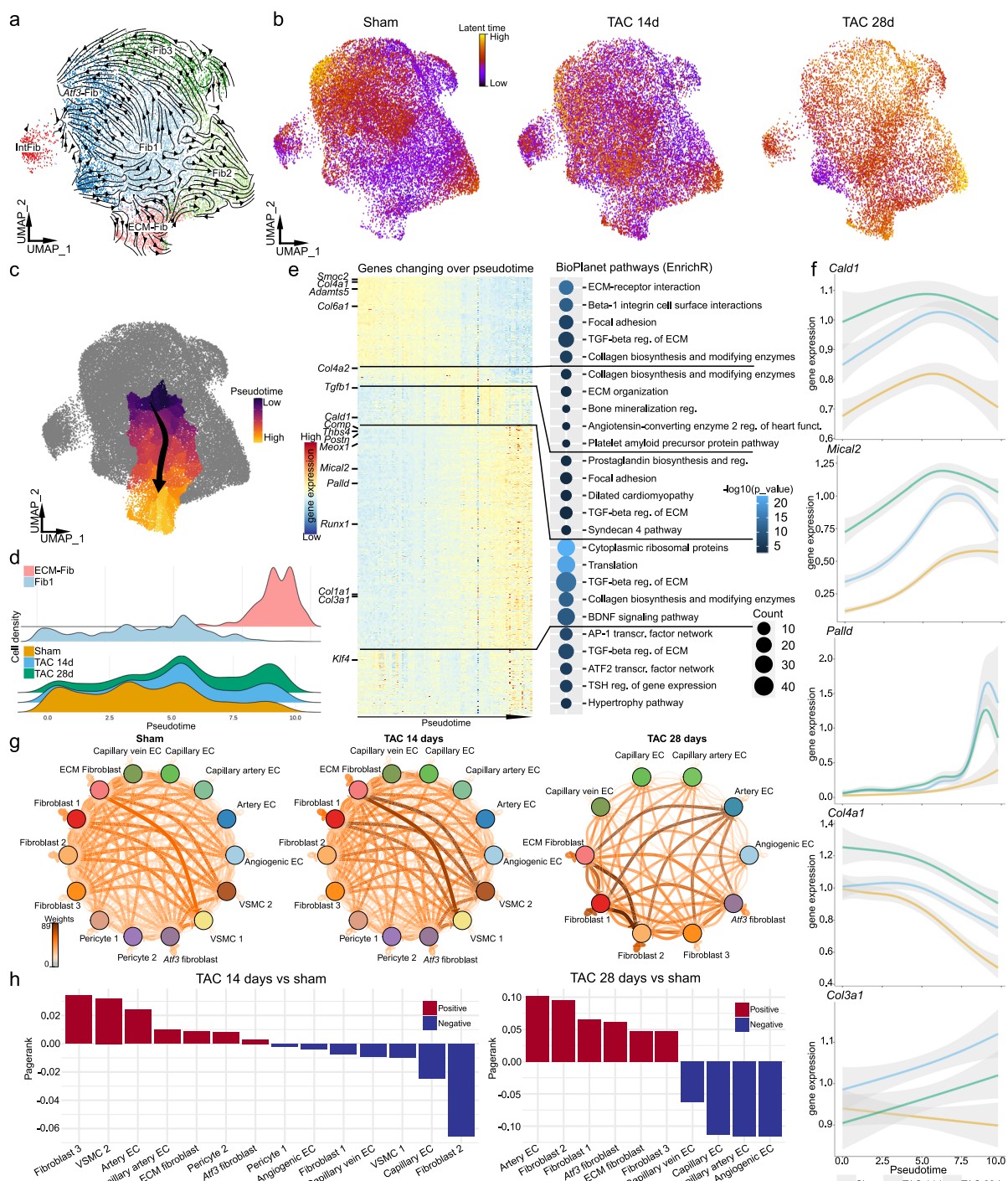

**Fig. 5 Analysis of potential fibroblast differentiation trajectories. a** Velocities derived from the dynamical model for subclustered fibroblasts (TAC day 28) visualized as streamlines in a UMAP embedding. **b** UMAP embedding of fibroblast, separated per condition with calculated latent time (from scVelo[113]). **c** Supervised trajectory from Fib1 to ECM-Fib estimated with Monocle 3. Color indicates calculated pseudotime. **d** Density plots for cell density per pseudotime of different groups of fibroblasts. Upper panel, Fib1 and ECM-Fib density per pseudotime, lower panel density of all fibroblasts separated by condition. **e** Heatmap of genes with a changing expression pattern over the calculated pseudotime in **c** and associated pathway activities. The heatmap was clustered into five gene clusters by hierarchical clustering and the gene clusters were sorted according to pseudotime. Enriched pathways were mapped with EnrichR[103] using Bioplanet[65] pathways database as a resource. Dot color indicates significance (−log10(p value)), dot size refers to overlap of tested genes and gene set (count). **f** Smoothened gene expression for *Cald1*, *Mical2*, *Palld*, *Col4a2* and *Col3a1* over pseudotime (*x*-axis) separated by condition: Sham, TAC 14 days and TAC 28 days (gray error band: 95% confidence interval). **g** Ligand receptor interactions based on CrossTalkeR between selected subclusters defined in this study separate for each time point. Color indicates weight of interaction. **h** Bar graph showing changes in interactions, comparing sham and TAC (left TAC 14 days, right TAC 28 days). Increase in interactions indicated in red, decrease in blue. For details on statistics and reproducibility, see "Methods".

specifically differentiated subtypes[48]. Col1a15 was found as the highest enriched marker gene for Fib1 (Fig. 3d), providing further evidence for Fib1 as ECM-Fib precursor. As ECM-Fib continuously increased in number after TAC (Fig. 3f), they potentially represent a more terminal differentiated subtype and therefore were selected as the endpoint of the trajectory. The calculated pseudotime of the trajectory path displays ECM-Fib enrichment toward later pseudotime (Fig. 5d), which is in line with the significant continuous expansion of this subtype after TAC (Fig. 3f). To understand the Fib1 to ECM-Fib differentiation trajectory we next sorted genes along this differentiation pseudotime by their expression change (Fig. 5e). Along the differentiation trajectory, early DEG were associated with ECM-receptor interaction and TGFβ signaling as well as focal adhesion, while late in this trajectory expressed genes were associated with dilated cardiomyopathy, syndecan 4 pathway, translation and the BDNF signaling pathway (Fig. 5e). A unique group of genes with a high expression profile at the start and end of the trajectory, was associated with the AP-1 and ATF2 transcription factor network and hypertrophy pathway (Fig. 5e). Looking into individual genes, we found TEAD1 activity associated genes *Cald1*, *Mical2* and *Palld* mid-to-late expressed in pseudotime (Fig. 5f). Interestingly, we observed a high number of ECM related genes changing expression over pseudotime with a shift in the expression of collagen subtypes from network collagens to fibrillar collagens (Fig. 5e, f and Supplementary Fig. 7f, g). This indicates a strong change in the ECM related expression profile of Fib1 while differentiating into ECM-Fib. Of note, the differentiation trajectory described above should be interpreted as a model, which is supported by our data.

**Cellular crosstalk of the vascular niche**. To understand the cellular receptor-ligand interactions within the vascular and perivascular niche we utilized CrossTalkeR[49]. Here, we focused our analysis on selected subtypes of the niche and excluded clusters containing low cell numbers and clusters with minor transcriptional changes after TAC. Interestingly, at day 14 and 28 after TAC we observed heavy crosstalk between ECM-Fib and Fib1 and other cardiac cell types such as VSMC and endothelial cells (Fig. 5g–i). Since ECM-Fib were the major fibrosis driving cells in our dataset and also showed the heaviest crosstalk after injury, we next focussed on this fibroblast type. PI3K-AKT signaling and focal adhesion related genes were highly enriched in crosstalk pathways of ECM-Fib with other cell types at day 14 after TAC as compared to sham (Supplementary Fig. 7h). At 28 days after TAC, ECM-Fib crosstalk with other cardiac cell types involved these two pathways and also ECM receptor interaction, MAPK, RAP1, RAS and Relaxin signaling (Supplementary Fig. 7h). PI3K-AKT has a known key role in angiogenesis[50] and the focal adhesion complex is the major interactor with the ECM. We already described changes in ECM expression by fibroblasts and it is likely that EC sense these changes in ECM composition involving the focal adhesion complex[51]. Therefore, we next turned toward the endothelium to understand heterogeneity and transcriptomic changes of this key vascular cell type in cardiac homeostasis and HF.

**Endothelial heterogeneity in murine heart**. Changes in endothelium function or even loss are known to negatively impact HF disease progression[52]. Since *Cdh5CreER* did specifically label the cardiac endothelium (Fig. 2a–d), we now focused on analyzing the single-cell gene expression data generated from these mice separately. Stringent filtering resulted in a total of 14,595 high quality EC transcriptomes for analysis (Fig. 6a and Supplementary Fig. 8a). We identified the following endothelial subtypes:

capillary EC (CapEC), capillary artery EC (CapA-EC), capillary vein EC (CapV-EC), stressed EC (StrEC), angiogenic EC (AngEC), interferon EC (IntEC), artery EC (ArtEC), lymphatic EC (LymEC), cycling EC (CyclEC), DNA replicating EC (RepEC) and endocardial EC (EndoEC). Our annotation was based on the top 20 marker genes per endothelial subtype specifically for heart and is consistent with the extensive murine EC atlas from ref. [53] (Fig. 6b). Most EC clusters in our dataset were easily identified based on this published atlas, however, we noticed differences for some populations (Fig. 6b). StrEC were found to be associated with response to stress and showed significant enrichment for stress induced genes like *Hspa1a*, *Fos* and *Fosb* (Fig. 5c, d and Supplementary Data 1). The clusters RepEC and CyclEC were annotated according to their high DNA synthesis (S) phase score in RepEC, high G2M phase scores in CyclEC (Supplementary Fig. 8b, c) and their marker genes (*Mcm3, Mcm5, Top2a*; Fig. 6d). For the EndoEC population, we observed a distinct expression of *Npr3*, a specific endocardial marker gene (Fig. 6e, f)[54]. Additionally, we found a high expression of *Vwf*, a well-known broad marker of EC, also expressed in the endocardial layer[55].

**No evidence for endothelial cell contribution to the mesenchymal lineage after TAC**. Interestingly, endothelial cells appear to have the potential to upregulate collagen expression during cardiac remodeling and have been suggested as a major source of fibrosis driving cardiac myofibroblasts via endothelial mesenchymal transition (EndoMT)[11]. However, in our dataset all *Cdh5* fate traced cells showed tremendously lower ECM gene expression as compared to all fibroblast populations and also pericytes (Fig. 3a and Supplementary Fig. 3e). Furthermore, the overall ECM gene expression in these cells did not increase substantially after TAC (Supplementary Fig. 3e). We did not observe strong similarities of *Cdh5* fate traced cells with the fibroblast clusters in the full integration of all mice since no *Cdh5*-tdTom[+] cell was observed close to the fibroblast populations (Fig. 2c, d). Similarly, no fibroblasts were found in the *Cdh5* fate traced populations (Fig. 6a–d). We further did not observe enrichment of a recently reported transient mesenchymal activation (EndMA)[10] gene set in endothelial cells after TAC (Supplementary Fig. 8d). Taken together these findings indicate no evidence for EndoMT/EndMA in the context of our 4-week TAC model.

**Angiogenic endothelial cells expand in response to cardiac remodeling**. Composition analysis showed an increase in CyclEC and RepEC numbers at day 14 after TAC, with a subsequent decrease of both populations at the later time point at even significantly lower numbers as compared to sham (Fig. 6g and Supplementary Fig. 8e). This suggests an initial pro-angiogenic phase of the injured heart in response to increased oxygen and metabolic demand[5], which is not sustained at the later time point. Quantification of Ki-67 tissue expression in *Cdh5* lineage tracing hearts confirmed this finding (Fig. 6h). RNA in situ hybridization experiments for the AngEC marker gene *Apln* showed enriched expression in a region of cardiac fibrosis, in line with an angiogenic response of EC particular in areas of injury (Supplementary Fig. 8f). Interestingly, while we observed a clear trend toward decreased capillary numbers this did not reach significance (Supplementary Fig. 8g). The observed decrease of proliferation EC after the initial phase might be related to the finding of a recent study, which demonstrated an abortive angiogenic response in the heart[56]. We also observed a loss of EndoEC and LymEC in our composition analysis (Fig. 6g). The loss in EndoEC can be explained by EndoEC damage as a result of the stress induced on the heart, which was demonstrated in a rat HF model[57]. The decreasing proportion of LymEC could have

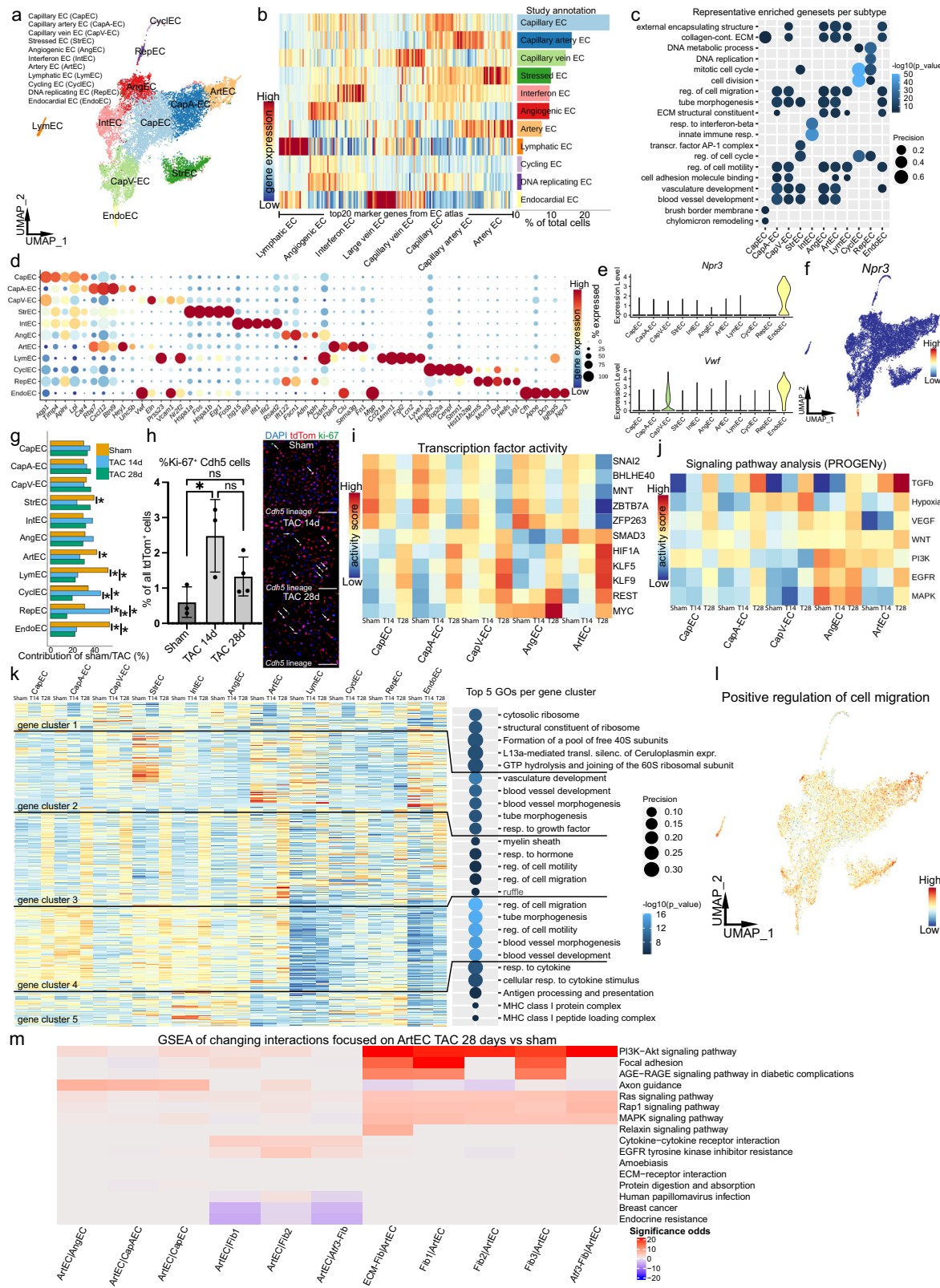

influenced HF progression in our model. This is described in a recent preprint reporting poor cardiac lymphangiogenesis accelerating HF development[58].

**Cardiac injury induces hypoxia related regulatory programs in EC.** We next analyzed our EC data for changes in transcription factor and signal pathway activity after TAC. For both analyses

we focused on CapEC, CapA-EC, CapV-EC, AngEC and ArtEC. TF activity prediction analysis indicated an increased activity in HIF1α, KLF5, KLF9, REST, and MYC, which was most prominent in ArtEC (Fig. 6i and Supplementary Fig. 9a). KLF5 has complex regulatory functions in endothelial cells, which depending on specific conditions, include proliferation and pro-inflammatory signaling[59]. Interestingly, KLF5 overexpression is

**Fig. 6 Exploration of TAC induced effects on cardiac EC. a** UMAP of 14,595 subclustered *Cdh5CreER;tdTomato* fate traced endothelial cells (EC) after integration from all conditions. Cluster annotation: Capillary (CapEC), capillary artery (CapA-EC), capillary vein (CapV-EC), stressed (StrEC), angiogenic (AngEC), interferon (IntEC), artery (ArtEC), lymphatic (LymEC), cycling (CyclEC), DNA replicating (RepEC) and endocardial (EndoEC). **b** Left, gene expression heatmap of 20 marker genes per EC atlas subtype[53] (x-axis). Right, *Cdh5* lineage derived dataset annotation and subcluster proportion. **c** Top five representative gene sets enriched (hypergeometric test, one-sided) in marker genes per EC subtype. Dot size refers to overlap of tested genes and gene set (precision). Full list in Supplementary Data 1. **d** Top five expressed marker genes per cluster. Dot size refers to proportion of cells expressing the gene per cluster. **e** Violin plot of *Npr3* and *Vwf* expression in EC subtypes. **f** UMAP visualization of *Npr3* expression. **g** Normalized proportion of EC per subcluster (*: false discovery rate (FDR) <0.05 and absolute log2 fold change >0.58, see also Supplementary Fig. 8d). **h** Cardiac Ki-67 quantification, values are expressed as percentage Ki-67+-tdTom+ of all tdTom+ cells (mean ± SD; n = 3 sham, n = 3 TAC 14 days, n = 4 TAC 28 days, independent replicates; *p value <0.05; one-way ANOVA). Source data are provided as a Source Data file. Right, representative images (scale bar 50 μm). **i** TF activity prediction based on TF regulons (DoRothEA) for selected EC subtypes. **j** Signal pathway activity prediction of selected EC subtypes based on pathway responsive genes (PROGENy). **k** Gene expression heatmap of hierarchical clustered, differentially expressed genes (DEG) combined with gene set enrichment analysis (GSEA). DEG were identified using MAST (only upregulated genes with adjusted p value <0.01 and logFC >0.3). Right, top five enriched gene sets (hypergeometric test, one-sided) per gene cluster. Dot size refers to overlap of tested genes and gene set (precision). **l** UMAP visualization of positive cell migration regulation scores. **m** GSEA of changing interactions between sham and TAC 28 days involving ArtEC. Color indicates up/down significance odds. For details on statistics and reproducibility, see "Methods".

described to compromise microvascular EC proliferation[60]. HIF1α and MYC are closely related transcription factors under hypoxic conditions[61]. In contrast to the predicted activity, we found a lack of *Hif1α* expression in ArtEC, while HIF1α induced genes known to be expressed in hypoxia, e.g., *Eno1*[62], were clearly expressed (Supplementary Fig. 9b). Interestingly, HIF1α has also been described as a negative regulator of TGFβ signaling in TAC[63] and we observed decreased SMAD3 TF activity. This is in contrast with the predicted increase in TGFβ pathway activity in ArtEC, potentially by non-canonical activation (Fig. 6i, j)[64]. SNAI2 was reported as an important positive regulator of angiogenesis[65] and showed decreased activity particularly 28 days after TAC, in line with reduced angiogenesis at this time point as described above (Fig. 6g, h). Signal pathway activity prediction indicated increased TGFβ and hypoxia pathway activity (Fig. 6j and Supplementary Fig. 9c). Additionally, VEGF, WNT, EGFR and MAPK signaling pathways were increased particularly in AngEC following TAC (Fig. 6j and Supplementary Fig. 9c). VEGF is a known master regulator of angiogenesis[66] and direct target of HIF1α[67].

In order to identify other injury-related transcriptomic changes in EC, we next performed differential gene expression analysis, gene clustering and GSEA (Fig. 6k). Overall, there were more pronounced changes at the 28 day time point (Supplementary Fig. 9d). The DEG were in a large part associated with regulation of cell migration and motility (gene cluster 3 and 4), as well as blood vessel developmental and growth factor response (gene cluster 2). Since migration is essential for angiogenesis[68], we next compared the different EC populations in regard to expression of a gene set of positive cell migration regulators and detected increased expression following TAC particularly in ArtEC (Fig. 6l and Supplementary Fig. 9e). Since we observed prominent transcriptomic changes of ArtEC as well as strong alterations of their cellular interactions (Fig. 5h), we focused our ligand receptor (LR) interaction analysis on this population (Fig. 6m and Supplementary Fig. 9f). This analysis suggested increased PI3K-AKT, MAPK, RAS, and RAP1 signaling between ArtEC and fibroblasts (Fig. 6m). Intracellular signaling involving MAPK has been described to be involved in cell migration[69] and can even be activated by mechanical forces[70]. The transcriptional changes particularly detected in ArtEC can potentially be explained by direct effects of the TAC model on coronary arteries, where hemodynamic changes induce endothelial damage followed by fibrotic remodeling[71]. The significantly diminished proliferative EC populations at 28 days after TAC together with the detected transcriptional changes at this time point could contribute to the mismatch in capillary density to cardiomyocyte size, which is

thought to be involved in cardiac hypertrophic remodeling and HF[72,73].

**Pericytes contribute to increased ECM deposition in cardiac remodeling.** Mural cells of the vascular wall consist of pericytes and VSMC. Cellular heterogeneity of these two cell types and their close relation have become more into focus of recent research[3,74], however their exact role and heterogeneity during HF remains mostly unknown. We were able to cover 7309 mural cells by lineage tracing derived from *Myh11CreER*, *Ng2CreER* and partially from *PdgfrβCreER* mice (Fig. 2a–f and Supplementary Fig. 10a). We excluded *Gli1* lineage derived mural cells from analysis, as the number of cells from the Cre driver was extremely low (Fig. 2c, d). We identified eight distinct subclusters within this mural population (Fig. 7a). Based on marker gene expression and related enriched gene sets we defined three VSMC and four pericyte subclusters (Fig. 7b–d and Supplementary Data 1). Beside classical mural cells, we also found a cluster of Schwann cells (Sw). These cells were mostly derived from the *Ng2* lineage (Fig. 6e). Besides being a well-known marker for pericytes in the adult heart and other organs[74], NG2 is known to be expressed in the adult central nervous system explaining its name neural glial antigen 2[75]. We observed no significant transcriptomic changes in the Sw cluster after injury (Supplementary Fig. 10b). The VSMC subtypes 1 and 2 were transcriptionally highly similar and differed mostly by different levels of contractile gene expression, with VSMC2 expressing less *Acta2*, *Tagln* and *Myh11* (Fig. 7b). We found a subcluster of stressed VSMC and stressed pericytes, which showed beside their cell type specific marker gene expression similar gene expression profiles associated with the AP-1 complex and stress response (Fig. 7d). Similar to *Atf3*-Fib, we observed minor *Atf3* expression also in mural cells (Supplementary Fig. 10c). Pericyte 1 (Peri1), pericyte 2 (Peri2) and interferon pericytes (IntPeri) expressed described pericyte markers *Kcnj8*, *Abcc9*, *Colec11*[76] (Fig. 7b). The interferon response signature of IntPeri was similar to the expression signature in IntFib and IntEC (Figs. 3d and 6d). We validated the presence of *Ifit1* expressing pericytes byin situ hybridization (Supplementary Fig. 10d). Peri1 and Peri2 differed mostly by differences in expression levels of marker genes which were associated with more ECM interaction for Peri1 (Fig. 7b–d) and processes involving potassium channel regulation for Peri2 (Supplementary Fig. 10e). ECM related functions of Peri1 could potentially point toward a stronger functional focus for structural support of the microvasculature in this pericyte subtype[77], while the regulation of potassium channel activity was recently described to be important for the regulation of blood flow in the brain

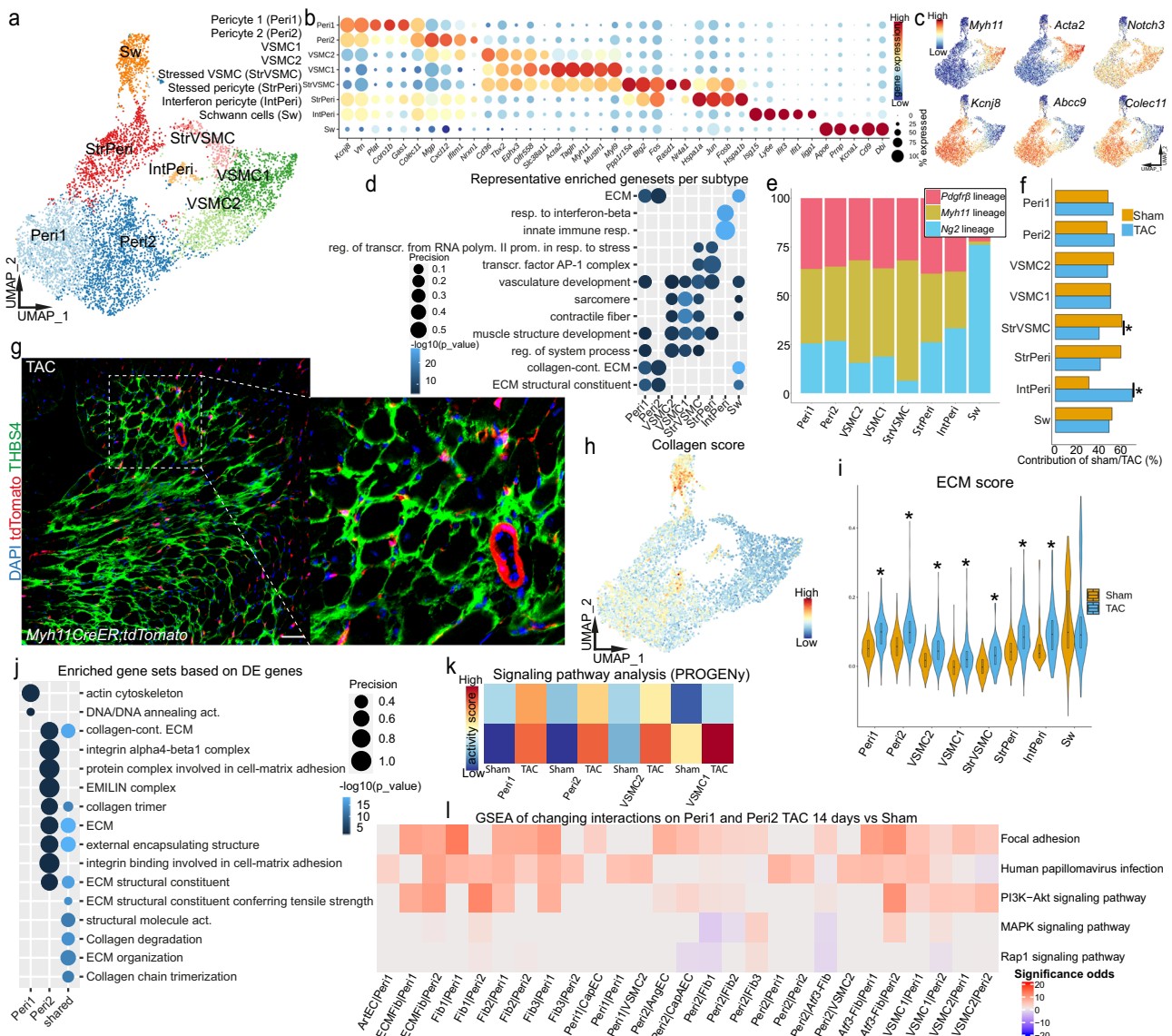

**Fig. 7 Mural cell heterogeneity and changes in TAC induced cardiac remodeling. a** UMAP of 7309 subclustered mural cells after integration of *PdgfrβCreER, Myh11CreER, Ng2CreER* derived cells from all conditions. Cluster annotation: pericyte 1 (Peri1), pericyte 2 (Peri2), vascular smooth muscle cell (VSMC 1), VSMC 2, stressed VSMC (StrVSMC), stressed pericyte (StrPeri), interferon pericyte (IntPeri) and Schwann cells (Sw). **b** Dot plot of top five expressed marker genes per cluster. Dot size refers to proportion of cells expressing the gene per cluster. **c** UMAP embedding showing gene expression of selected marker genes: *Myh11, Acta2, Notch3, Kcnj8, Abcc9, Colec11*. **d** Top two representative gene sets enriched (hypergeometric test, one-sided) in marker genes per mural subtype. Dot size refers to overlap of tested genes and gene set (precision). Full list in Supplementary Data 1. **e** Normalized contribution of each fate traced genotype to the corresponding cluster. **f** Normalized proportion of mural cells per subcluster from sham and TAC condition samples (*: false discovery rate (FDR) <0.05 and absolute log2 fold change >0.58, see also Supplementary Fig. 10f). **g** Confocal immunofluorescence image of *Myh11CreER* tagged TAC heart stained for THBS4 (scale bar 50 µm). **h** UMAP embedding with ECM score per cell. **i** Violin plot of summarized ECM scores per condition and subcluster (all three fate traced fibroblast genotypes combined). Integrated boxplots show center line as median, box limits as upper and lower quartiles. Peri1 and Peri2 show a significant difference between sham and TAC (*p < 0.001, two-sided Wilcoxon rank sum test, unpaired, Bonferroni adjusted p value). **j** Top significant enriched gene sets associated with either shared or unique differentially expressed genes (DEG) per subtype. Dot size refers to overlap of tested genes and gene set (precision). **k** Signal pathway activity prediction of selected EC subtypes based on pathway responsive genes (PROGENy). **l** Gene set enrichment analysis (GSEA) of changing interaction involving Peri1 and Peri2. Color indicates up- or down significance odds. For details on statistics and reproducibility, see "Methods".

microvasculature[18]. Comparison of mural subtype contribution per condition to each subcluster revealed a significantly higher proportion of IntPeri in TAC (Fig. 7f and Supplementary Fig. 10f). In line with this observation, proliferative expansion or substantial loss of mural cells was not recognized by confocal imaging following TAC (*Myh11CreER;tdTomato*, Fig. 1f and Supplementary Fig. 1e). Since THBS4 was a good marker for regions of injury associated ECM-Fib, we performed

THBS4 staining on tissue sections from our *Myh11* and *Ng2* fate tracing hearts to be able to specifically focus on areas of interstitial fibrosis (Fig. 7g and Supplementary Fig. 10g). We were not able to observe any expansion or loss of tdTom⁺ cells in these fibrotic THBS4⁺ regions.

After applying the ECM scoring to the mural cell subclusters, we observed a higher collagen score for pericytes compared to VSMC, as well as an overall higher ECM score (Fig. 7h, i). This

score was higher in cells from the TAC condition, indicating that pericytes start producing collagen similar to fibroblasts. However, we need to point out that the total ECM score of mural cells compared to fibroblast populations was clearly lower (Supplementary Fig. 3e). In line with this finding, differential gene expression analysis detected collagens as highest upregulated genes in TAC for Peri1 and Peri2 (Supplementary Fig. 11a, b). Overall pericytes showed a stronger reaction to TAC with more genes being up and downregulated compared to VSMC (Supplementary Fig. 10b). GSEA of DEG also suggested mainly terms associated with increased ECM expression in both pericyte subtypes (Fig. 7j). We next performed RNA in situ hybridization to validate pericytes as contributors to matrix production after injury. Co-staining of the pericyte marker *Kcnj8* with *Col1a1* in injured hearts identified only a very low number of *Col1a1* co-expressing *Kcnj8*[+] pericytes (Supplementary Fig. 11c), suggesting that pericytes might not contribute much to fibrillar matrix expression. Analysis of pathway activity focused on the major pericyte and VSMC clusters revealed strong association with a specific upregulation of TGFβ and JAK-STAT signal pathway activity (Fig. 7k and Supplementary Fig. 11d).

In our subsequent analysis for changes in receptor-ligand interaction in mural cells, we focused on signaling from and to Peri1 and Peri2, as these cells showed the most transcriptomic changes (Fig. 7l). Cellular crosstalk between pericytes and endothelial cells have been implicated to have an important role in angiogenesis[78,79]. This analysis revealed changes in signaling between pericytes, endothelial cells and fibroblasts, involving notch signal pathway, PI3K-AKT and focal adhesion (Fig. 7l). Overall, the results suggest a role of pericytes in the regulation of angiogenic processes in the context of cardiac remodeling.

## Discussion

We have here combined inducible genetic fate tracing with high-resolution confocal imaging and scRNA-seq to dissect changes and heterogeneity in the vascular and perivascular niche during HF. Fate tracing experiments using six different Cre drivers revealed changes in localization and transcriptome of fibroblasts, endothelial and mural cells in an early phase of hypertrophic cardiac remodeling.

Integration of scRNA-seq data obtained from these fate tracing experiments clearly demonstrated which cell types are labeled by each Cre driver and the plasticity of endothelial, mural and fibroblast populations in pressure overload induced cardiac remodeling. *PdgfrβCreER*, *Gli1CreER* and *Col1a1CreER* all recombined within all fibroblast subtypes with the latter two being fibroblasts specific and *PdgfrβCreER* being the broadest mesenchymal Cre driver tested that also recombined in mural cells, in line with previous studies[80,81]. As expected *Cdh5CreER* was EC specific, while *Myh11CreER* and *Ng2CreER* specifically labeled mural cells. It should be noted that *Ng2CreER*, albeit being mural cell specific, was significantly less efficient for labeling cardiac mural cells as compared to *Myh11CreER*. This suggests that *Myh11CreER* is a better tool for further lineage tracing studies of cardiac mural cells.

We have previously reported that *Gli1* marks cardiac myofibroblast precursors that possess mesenchymal stem cell (MSC) characteristics in the dish and that genetic ablation of these cells ameliorates cardiac fibrosis and stabilizes cardiac function after ascending aortic constriction[82]. Our scRNA-seq data here indicates that *Gli1CreER* mice recombine in the various cardiac fibroblast populations. This raises the question whether all cardiac fibroblasts would possess potential MSC characteristics in the dish and whether a difference between fibroblasts and MSC-like cells exists.

A recent study by Soliman et al. reports a distinct heterogeneity of cardiac fibroblast, where a large population of PDGFRα[+]/SCA-1[+] cells display MSC-like progenitor features[47]. These progenitors generate PDGFRα[+]/SCA-1[−] fibroblasts upon injury. Our single-cell data identified two different fibroblast populations with enriched *Sca-1* (gene symbol: *Ly6a*) expression, Fib2 and IntFib. Both subtypes seem to be conserved independent of injury models as they were also found by refs. [15,23]. In contrast to the loss of PDGFRα[+]/SCA-1[+] fibroblasts after injury as reported by Soliman et al., *Sca-1* high Fib2 did not decrease after injury in our study and others[15,23]. Interestingly, Tang et al. reported that *Sca-1* fate traced PDGFRα[+] cells expand in the injury region after MI[83]. Our trajectory interference analysis suggested a more general pool of fibroblasts as potential progenitors, mostly consisting of the Fib1 type, however, this computational model needs to be considered carefully. The progenitor role of cardiac fibroblasts remains controversial and more experiments are needed to dissect the progenitor potential of the different cardiac subtypes.

Our data confirms that fibroblasts are the major source of fibrosis in the murine heart following pressure overload[4]. Furthermore, all fibroblast subtypes seem to get activated and increase their ECM gene expression contributing to scar formation, however we identified ECM-Fib as the highest ECM gene expressing population. We think subtypes similar to ECM-Fib were found by previous single-cell sequencing studies[15,23]. McLellan et al. describe FibThbs4 and FibCilp as injury-related subtypes with increased ECM remodeling related gene expression. Forte et al. found a fibroblast subtype that occurred only at later stages of cardiac remodeling 14 days after MI, which they termed MFCs according to an earlier described fibroblast type[84], found in the maturing scar. Taken together, it seems plausible that these results all describe a similar subtype of fibroblasts, which occurs in non-acute cardiac remodeling and is characterized by higher expression of ECM related genes.

We did not observe specific expression of *Acta2* for the ECM-Fib subtype or any other fibroblast subtype in our study even when integrating with data from other murine heart disease models[15,23], indicating that in our injury model no classic myofibroblasts developed. We propose that depending on the type of cardiac injury the requirement of contractile proteins is needed in scar forming myofibroblasts. This might be particularly important following MI to quickly close the large defect of the left ventricular wall in replacement fibrosis while in interstitial fibrosis following pressure overload this might not be an important feature of myofibroblasts. For many years the term myofibroblast was used for the major matrix producing fibrosis driving cell types. However, data presented here and various other scRNA-seq datasets across major organs[15,19,23,84,85] indicated that the major matrix producing cell types are often not expressing classical myofibers (*Acta2*) and thus the terminology needs to be adapted.

Comparison of DEG of all fibroblast subtypes revealed a general transcriptional change including upregulation of *Postn*. *Postn* was previously described as a myofibroblast marker[86]. Our data suggest that *Postn* expression increases in all fibroblast clusters including the ECM-Fib population, suggesting *Postn* as a broad marker of fibroblasts activation. In TAC hearts, *Thbs4* specifically marked the ECM-Fib population which was defined by highest ECM expression in our dataset and we observed extensive expansion of these cells in focal fibrotic areas of the myocardium. While ECM-Fib were also present in homeostasis, the lack of gene expression and presence of THBS4 in areas of homeostatic hearts indicates this subtype is specifically activated after injury. Our data indicated that TEAD1 is strongly activated in ECM-Fib after injury and in vitro validation confirmed that TEAD1 overexpression can induce fibroblast activation and matrix expression

in a Gli1[+] cell-line. TEAD1 is an important cofactor of the hippo pathway and it has been shown that *Tead1* expression correlates with expression of *Acta2* in murine fibroblasts where the *Tead1* motif was enriched particularly post MI[41]. Several recent studies focusing on the hippo pathway and its central downstream TF YAP, describe a complex regulatory role in wound healing and scar formation by regulating fibroblasts activation[87], cardiomyocyte proliferation and differentiation[39,88].

Regulation of angiogenesis is an important process in cardiac remodeling and increased angiogenesis prevents functional decline in HF[89]. Increased angiogenesis occurs early during cardiac remodeling while it later transitions into capillary dysfunction and capillary loss[90]. The captured 2-week time point in our study likely results from the early acute phase of compensatory mechanisms, in line with the observed expansion of the RepEC and CyclEC populations, but is not maintained until the 4-week time point. Proliferation of EC from pre-existing vessels has been shown to be the major source of cardiac neovascularization[54]. However, the poor pro-angiogenic potential of cardiac EC has previously been demonstrated[56] and we found several changes which might be responsible, including a mixture of pro- and anti-angiogenic TF and pathway activities. Especially ArtEC seem to react to TGFβ, which has been described as an anti-angiogenic factor[89]. Changes in cellular interaction were highly related to ECM-Fib and ArtEC including PI3K-AKT signaling and focal adhesion complex interactions. These functional complexes likely play a central role in the niche interaction changes, since the ECM composition is changed by the ECM-Fib, which in turn can be sensed by the ArtEC. Interestingly, targeting integrin subtypes, which are the connection between the focal adhesion complex and the ECM, has already been shown as promising therapeutic targets in fibrotic remodeling[27]. The cellular origin of cardiac myofibroblast has been debated for a while[91] and even endothelial cells via EndoMT have been suggested as a major contributor[11]. The plasticity of endothelial and mesenchymal cells during disease has been demonstrated in several publications[7–11]. Importantly, our combined inducible genetic fate tracing and scRNA-seq data clearly points against plasticity of endothelial and mesenchymal cardiac cells in murine pressure overload induced HF.

Mural cells, as one of the major non-myocyte cell types, have become more into the focus of cardiac research[92]. We demonstrate here an unprecedented map of mural cells in HF. Similar to recently published data[3], mural cells seem less heterogeneous than fibroblasts or EC. Interestingly, we detected a transcriptional intermediate stage between pericytes and VSMC, which has been reported recently[3]. Pericytes have been described as progenitors of coronary artery smooth muscle cells[92]. Our data demonstrates a close transcriptional relationship of these cell types, while we did not observe previously reported potential differentiation processes toward other cardiac lineages[93]. However, we observed that mural cells increase the expression of ECM related genes after injury. Especially pericytes express more collagens, which might be explained by the structural supportive function of pericytes toward EC in small capillaries[48]. Since we were only able to generate a single early time point after TAC and no functional validation for mural cell contribution to cardiac scar formation, further studies are needed to validate their contribution to cardiac remodeling and fibrosis. Furthermore, we need to point out that we lack the acute phase directly after TAC surgery. In addition, the time frame of 4 weeks after TAC represents an earlier phase of hypertrophic remodeling and not the later decompensated phase of HF[94,95]. One potential limitation of our study is that in the *PdgfrβCre* and *Myh11Cre* crossings we did not see significant hypertrophy suggesting a milder phenotype and or variance in the model. Additionally, we acknowledge that our lineage tracing based enrichment strategy excluded cardiomyocytes, therefore we lack analysis of the potential influence from this cell population. Lastly, trajectory analyses and cell-cell communication analyses represent computational predictions, thus interpretation must be considered with caution.

Taken together, our study presents insights into the transcriptomic changes of the heterogeneous perivascular niche of the murine heart in hypertrophic remodeling. We identified a specific fibroblast subpopulation that exists during homeostasis acquires *Thbs4* expression and expands after injury. Endothelial cells showed a proliferative response at our first time point, which was not sustained in later remodeling, together with transcriptional changes related to hypoxia, angiogenesis, and migration.

## Methods

**Ethics**. The local ethics committee of the University Hospital RWTH Aachen approved all human tissue protocols (EK 151/09).

**Animals**. All animal experiment protocols were approved by the LANUV-NRW (protocol number 81-02.04.2018.A020) Germany. *PdgfrβCreER*[t2] (i.e.,B6-Cg-Gt(Pdgfrβ-CreERT2)6096Rha/J], JAX Stock #029684; $n = 9$, 2 male, 7 female), *NG2CreER* (B6.Cg-Tg(Cspg4-Cre/Esr1*)BAkik/J], JAX Stock #008538; $n = 6$, 6 female), *Gli1CreER* (Gli1tm3(Cre/ERT2)Alj/J], JAX Stock #007913; $n = 14$, 7 male, 7 female), *Myh11CreER* (B6.FVB-Tg(Myh11-Cre/ERT2)1Soff/J], JAX Stock #019079; $n = 6$, 6 male) were purchased from Jackson Laboratories (Bar Harbor, ME, USA). C57Bl6/129SV-Collagen1alpha1-GFP-CreERT2 (*Col1a1CreER*, $n = 10$, 4 male, 6 female) was a kind gift of Ivica Grgic (Marburg). Cdh5(PAC)-CreERT2 (*Cdh5CreER*, Taconic no. 13073; $n = 12$, 4 male, 8 female) was a kind gift of Rui Benedito (Madrid). All Cre driver lines were crossbred with *Rosa26tdTomato* (i.e., B6.Cg-Gt(ROSA)26Sor[tm9(CAG-tdTomato)Hze]/J, JAX Stock #007909) also from Jackson Laboratories, to generate the according tamoxifen inducible fate tracing genotype. Mice were housed with two to five animals per cage at a 12 h light-dark cycle at sustained temperature (20 °C ± 0.5 °C) and humidity (~50% ± 10%) and ad libitum access to food and water. For inducible fate tracing mice (8 weeks of age) of all different strains were injected intraperitoneal four times with 3 mg tamoxifen. After a washout period of 21 days, TAC was performed to induce cardiac hypertrophy or sham surgery as a control procedure. In brief, mice were anesthetized and analgesia applied. After intubation, the chest was opened via the second intercostal space at the left upper sternal border through a small incision and aortic constriction was performed by tying a ligature against a 27G needle. Control mice underwent a sham operation with a skin incision. All mice were sacrificed 14 or 28 days after surgery.

**Human tissue sample**. The human cardiac specimen was obtained from a 50-year-old patient who underwent left ventricular assist device surgery due to ischemic cardiomyopathy. The local ethics committee of the University Hospital RWTH Aachen approved all human tissue protocols (EK 151/09). All tissue samples used for this study were obtained with written informed consent from all patients in accordance with the guidelines of The Declaration of Helsinki 2000.

**Echocardiography measurement**. Echocardiography was performed on a Visualsonics Vevo 3100 (FUJIFILM VisualSonics, Canada) in 2% isoflurane anesthesia. Respiration rate and heartbeat were continuously monitored throughout via the stage electrodes. Thoracic fur was removed by depilatory cream and mice were fixed on a heated table (37 °C). The MX550D echocardiography transducer was used to scan the parasternal long and short axis views in B-Mode, with a target heart rate of 450–500 bpm. The peak trans-TAC pressure was measured as previously described[95]. In brief, color doppler was used to visualize flow at the constriction and the pulse-wave was measured by doppler. The peak pressure was calculated using the Bernoulli equation. To acquire 4D imaging data, the step motor was positioned just below the apex and the motor aligned to take concentric short axis images in 0.2 mm steps. At each position, a complete cardiac cycle was recorded using automated ECG and respiratory gating. Ejection fraction was calculated based on volumetric measurement of the 4D images using Vevo Lab (v5.5.1, FUJIFILM VisualSonics, Canada).

**Single-cell isolation and FACS sorting**. Mice were euthanized and the heart was perfused with 20 ml PBS (Gibco™ PBS, pH 7.4, 10010056). Hearts were removed, weighed and kept on ice until all hearts were collected for tissue dissociation. To generate a single-cell solution, hearts were cut transversely where upper half was used for imaging and lower halves were minced in a petri dish on ice using a scalpel, and transferred to C-tubes (gentleMACS C Tubes, 130-096-334) containing with 6 ml digestion medium (RPMI 1640 #31870025 with 0.2 mg/ml Liberase™ TL Research Grade (Roche, 5401020001 and 60 U/ml DNase I (Sigma D5025)) and incubated for 15 min at 200 RPM at 37 °C on an orbital shaker. Afterwards C-tubes were run on the gentleMACS™ Dissociator using the spleen4 program followed by a

second 15 min 37 °C incubation on the orbital shaker. The sample was from now on always kept at 4 °C. Next, the sample was filtered through a 100 μm cell strainer and cells were washed by centrifugation for 5 min at 300 × *g* and resuspended in 20 ml ice cold FACS buffer (PBS + 2% fetal bovine serum). Erythrocytes were lysed for 5 min on ice (eBioscience 1X RBC Lysis Buffer, in MilliQ water) and the cells were washed again with 10 ml FACS buffer. For enrichment of lineage traced cells, the cells were stained with DAPI (Roche, 10236276001) and sorted a Sony LE-SH800 for DAPI negative, tdTomato mid-to-high signal (Supplementary Fig. 12). At least three individual hearts were pooled per condition and genotype.

**Fibrosis staining Sirius Red**. Fibrotic collagen matrix was stained using picro-sirius red and representative pictures imaged using brightfield microscopy (Leica).

**Immunofluorescence staining**. Extracted upper parts of the hearts were fixed for 2 h in 4% paraformaldehyde (Carl Roth) at RT. These parts were then transferred to 30% sucrose solution in PBS and kept at 4 °C overnight to prevent ice crystal formation. Samples were then embedded in Tissue-Tek O.C.T. Compound (Sakura) and quickly frozen and stored at −80 °C until sectioning. In total, 4 μm cryosections were then washed in PBS and permeabilized in PBST-0.1% (PBS with 0.1% Triton X-100 (Sigma-Aldrich). Blocking was performed in 10% BSA in PBS, followed by 1 h incubation of primary antibody. Slides were washed three times for 5 min in PBS and incubated afterwards with secondary antibody for 30 min. Lastly, nuclei were counterstained with 4′,6-diamidino-2-phenylindole (DAPI) staining (Roche, 1:10,000) and slides were mounted with ProLong Gold (Invitrogen, P10144). Used antibodies are listed here: Anti-Mouse CD31 (BD Biosciences, 1:100, 553370), Thrombospondin-4 Antibody (Novus Biologicals, 893655, 1:100) and Ki-67 Monoclonal Antibody (SolA15) (eBioscience, 14-5698-80, 1:100). AF488 donkey anti goat (Jackson Immuno Research, 1:200), and AF647 donkey anti-rabbit (Jackson Immuno Research, 1:200) and AF647 donkey anti-rat (Jackson Immuno Research, 1:200).

**RNA in situ hybridization staining**. RNA in situ hybridization was performed using OCT embedded tissue samples and the RNAScope Multiplex Detection KIT V2 (RNAscope, 323100, Advanced Cell Diagnostics, United States) according to the manufacturer's protocol. The following probes were used for the RNAscope assay: Mm-Apln 415371, Mm-Kcnj8 411391, Mm-Abcc9 411371, Mm-Iftit1 500071, tdTomato 317041, Mm-Postn 418587, Mm-Pdgfrα 480661, Mm-Colec11 855961, Mm-Alf3 4268691, Mm-Pdgfrα 480661, Mm-Pdgfrβ 411381, Mm-Col1a1 319371.

**Confocal imaging and Ki-67 quantification**. Images were acquired utilizing the Nikon A1R confocal microscope using ×10, ×40 and ×60 objectives (Nikon). Raw imaging data were processed using Nikon Software or ImageJ. Ki-67 stained hearts of *Cdh5CreER* lineage were analyzed using ImageJ, nuclei were assessed on tdTomato+ and Ki-67+ signal and quantified accordingly.

**Generation of murine cardiac Gli1+ fibroblast cell line**. *Gli1CreER;tdTomato* mice were sacrificed 2 weeks after tamoxifen treatment (Carbolution, CC99648). Cardiac tdTomato+cells were sorted by FACS and immortalized 14 days later with pBABE-puro SV40 LT (Addgene: #13970; a gift from Thomas Roberts). Retroviral particles were produced by transient co-transfection of HEK293T (ATCC, #CRL3216, Lot:70008735, aliquots from passage 2 were used for the experiments) cells with pCL-Eco (Addgene plasmid #12371, a gift from Inder Verma). In total, 72 h after transduction of Gli1+ cells, infected cells were selected with 7-day puromycin titration.

**Retroviral overexpression of murine Tead1 combined with TGFβ treatment**. For cloning of pMIG-mu*Tead1* the cDNA of *Tead1* was PCR amplified from murine heart tissue using the primer sequences 5′-CTAGATCTGCCACCATG GAGCCCAGCAGCTGGAG-3′ and 5′-ATACTCGAGTCAGTCCTTCACAAG CCTGTAGATATGG-3′. Subsequently, the PCR product was digested with BglII and XhoI and cloned into pMIG (pMIG was a gift from William Hahn (Addgene plasmid #9044). Retroviral particles were produced by transient transfection of HEK293T cells with the packaging plasmid pCL-Eco (Addgene plasmid #12371, which was a gift from Inder Verma) in combination with pMIG-mu*Tead1* or empty vector using TransIT-LT (Mirus). Viral supernatants were collected from HEK293T cells 48–72 h after transfection, clarified by centrifugation, supplemented with 10% FCS and Polybrene (Sigma-Aldrich, final concentration of 8 μg/ml) and 0.45 μm filtered (Millipore; SLHP033RS). Cell transduction was performed by incubating immortalized cardiac *Gli1CreER;tdTomato* labeled fibroblast cells with viral supernatants for 48 h. Transduction efficiency was analyzed by FACS-analysis for level of EGFP-expressing cells.

For TGFβ treatment, transduced cells were serum starved for 24 h followed by treatment with 10 ng/ml TGFβ (PeproTech, 100-21) or PBS control for 24 h. Cells were harvested and RNA isolated using the RNeasy Mini Kit (Qiagen, 74106) and reverse transcription performed (High-Capacity cDNA Reverse Transcription Kit, Thermo Fisher, 4368813). qPCR for *Tead1, Gapdh, Atca2, Col1a1* and *Fn1* was performed with SYBR Green Supermix (Bio-Rad, 1725125), using to following

primers: *Gapdh* forward primer (5′-AAGTGGTGATGGGCTTCCC-3′); *Gapdh* reverse primer (5′-GGCAAATTCAACGGCACAGT-3′); *Acta2*forward primer (5′-GTCCCAGACATCAGGGAGTAA-3′); *Acta2* reverse primer (5′-TCGGAT ACTTCAGCGTCAGGA-3′); *Col1a1* forward primer (5′-TGACTGGAAGAG CGGAGAGT-3′); *Col1a1* reverse primer (5′-GTTCGGGCTGATGTA-3′); *Fn1* forward primer (5′-ATCTGGACCCCTCCT-3′); *Fn1* reverse primer (5′-GCC CAGTGATTTCAG-3′); *Tead1* forward primer (5′-TCAAGCCGCCATTA AGGTGT-3′); *Tead1* reverse primer (5′-GCAGTAGCCGAGACGATCTG-3′).

**Single-cell RNA library generation**. In total, 16,000 FACS sorted cells were loaded for each sample (except for *Ng2*, ~2000 cells per sample) onto a Chromium Single-Cell B Chip Kit (v3.0, PN-1000073). cDNA libraries were generated following the standard protocol form Chromium Single-Cell 3′ GEM, Library & Gel Bead Kit v3 (PN-1000075) and sequenced on Illumina NovaSeq sequencer. Median reads per cell reached 35,220 with a median estimated saturation of 66.35%.

**Single-cell RNA data analysis**. All steps can be found in the corresponding R scripts (https://github.com/KramannLab/Murine_heart_map), including detailed description.

**Data processing and filtering**. Reads from the sequencing sequence were aligned to the mouse genome (modified mm10, including *tdTomato* sequence, see Supplementary Table 1) using cell ranger v3.0.2. Further analysis was performed using R v4.0.2 (https://www.r-project.org/) including several packages which are mentioned in the according section and listed in Supplementary Table 2. A schematic overview of the data analysis strategy is shown in Supplementary Fig. 2a. As an initial step, we filtered cells with less than 500–1000 features or more than 3000–5000 features, depending on the distribution of features per cell. For some samples we used a higher upper limit, as those samples showed an overall higher feature count. For all samples, cells with >6% of reads mapping to mitochondrial genes were removed. All cells containing reads for *Ptprc* (CD45), *Hba-a1, Hba-a2, Hbb-bs* (hemoglobins) were removed from further analysis, as those genes indicate contamination by immune cells and erythrocytes.

**Data integration of all datasets**. Harmony (v1.0)[12] was only used for the integration of all 14 datasets (Fig. 1 and Supplementary Fig. 2a, b). First all prefiltered datasets (see section: "Data processing and filtering") were merged into one Seurat object and based on this object, the data were normalized (NormalizeData from Seurat with default settings), variable features calculated ("vst" method), scaled (ScaleData from Seurat with default settings) and principal components calculated (RunPCA from Seurat, npcs set to 20). The RunHarmony function was run with default settings except for epsilon.cluster set to infinite.

Robust lineage tracing was one of the major prerequisites for this project. Therefore, filtering for cells and clusters displaying a clear expression for *tdTomato* on mRNA level was performed carefully. After the integration and batch correction by harmony, clusters were calculated using the FindNeighbors (reduction based on harmony, dimensions set to 1:20) and FindClusters (Louvain algorithm, resolution set to 1.0) function from Seurat. The probability calculation from genesorteR[96] was applied separately for each sample. Parts of a cluster were removed if the condGeneProb of the cells from an individual sample contributing to the cluster was below 0.8. All cells with 0 reads for *tdTomato* were also removed (Supplementary Fig. 2b).

As a further quality step, the number of features (genes) per cell were plotted as a violin plot grouped by cluster. We removed clusters with low feature count and/ or higher read fractions for mitochondrial and ribosomal genes. Two clusters showing mixed marker expression of mural cells, endothelial cells and fibroblast were removed, as they are potentially doublets (Supplementary Fig. 2b).

**Cell type specific data integration and further filtering**. For the focused analysis per major cell type (fibroblast, endothelial, mural cells) we performed integration based on Seurat method using canonical correlation analysis (Supplementary Figs. 2a, c, 8a and 10a)[97,98]. Following the standard integration workflow from Seurat, each prefiltered dataset (see section: "Data processing and filtering") was normalized (NormalizeData from Seurat with default settings) and the 2000 most variable features were calculated using the vst method. Integration features were selected with the FindIntegrationAnchors (using dims set to 1:20) function. *tdTomato* was manually excluded from the generated list of 2000 features. The datasets were then integrated with IntegrateData (using dims set to 1:20). After this integration, Clusters were calculated at a resolution of 0.5. Here we observed small clusters of zero to low *tdTomato* expression, which were not sufficiently excluded by the FACS sorting, probably due to doublets of a *tdTomato* positive and a *tdTomato* negative cell. For each cluster, the conditional probability of observing *tdTomato* (condGeneProb) was calculated with the sortGenes function from the package genesorteR v0.4.3[96]. Clusters with a low probability were removed from further analysis, as those clusters contained contaminating cells. All cells with 0 reads for *tdTomato* were removed afterwards. Clusters enriched for cells with relatively low feature count were identified as clusters of low quality cells, with higher read fractions for mitochondrial and ribosomal genes. These clusters were also removed from further analysis.

**UMAP**. All UMAP representations were calculated using the RunUMAP function from Seurat. To calculate the UMAP representation based on the Seurat integration, the following steps were applied. The data were scaled and principal components calculated with function from Seurat (default settings). Based on this, the RunUMAP function was applied with 30 PCA dimensions. For the UMAP representation using harmony, the RunUMAP function was applied with reduction set to harmony and dimension to 20.

**Annotation**. Cluster annotation for the full integration of all 14 samples was only applied at a low resolution level to identify major cardiac cell types covered by the lineage tracing. Subclusters of the major cell types were annotated based on information from literature, functional information from gene sets based on the marker genes of a subcluster and cell cycle status scoring.

**Cluster composition analysis**. Testing for significant changes in cluster composition regarding the contribution of the different conditions was performed by the scProportionTest package (https://github.com/rpolicastro/scProportionTest).

**Gene set enrichment analysis (GSEA)**. The gprofiler2 package[99] was used to perform GSEA based on marker genes and DEG. The gost function was run with default settings except for organism set to mmusculus and ordered query as true. Ribosomal genes were excluded from the gene lists beforehand and only gene lists with more than 5 remaining genes sorted by adjusted $p$ value were tested. Before plotting, the identified gene sets were filtered for terms with less than 1000 genes and only terms from GO[100–102], Reactome[100–102] and KEGG[100–102] were plotted. For testing pathway enrichment in the pseudotime associated genes, the enrichR package[103] was used with the BioPlanet 2019[104] database as reference.

**Marker gene identification**. Marker genes per subcluster were calculated by the Seurat function FindAllMarkers, using the method "Model-based Analysis of Single-cell Transcriptomics" (MAST)[29] and min-pct set to 0.3. Enriched gene sets per set of marker genes were analyzed as described in the GSEA section.

**Differential gene expression analysis**. DEG were calculated within each individual cluster, comparing expression differences between the different conditions with the FindMarkers function from Seurat, using the method MAST, including the following settings: min.pct = 0.25, logfc.threshold = 0.3. For all further analysis based on this result, only DEG with an adjusted $p$ value < 0.01 were included. For heatmap visualization of DEG, the average expression per gene and clusters was calculated with the AverageExpression function from Seurat. The resulting gene expression matrix was clustered by hierarchical clustering into 5–7 clusters (R package stats, default settings). Enriched gene sets per gene cluster were analyzed as described in the GSEA section. For volcano plot visualization, each gene was categorized if it was uniquely found to be differentially expressed in only one subcluster or if it was differentially expressed in two or more subclusters.

**Functional scorings**. In this study we performed multiple scoring approaches. Cells were scored using the AddModuleScore function from Seurat based on the following gene sets. Different ECM gene sets provided by the matrisome project[20]. Cell cycle scoring gene sets provided by Seurat with the CellCycleScoring function[105,106]. Different collagen subgroups as described by ref.[24]. GO term for positive regulation of cell migration GO:0030335. EndMA genes as published by ref.[10].

**Signal pathway and transcription factor activity**. We used the R package PROGENy v1.10.0[107,108] to estimate signaling pathway activities for each cell and summarized the scores for each subcluster per condition. In some cases the scoring was only applied to a subset of subclusters. Transcription factor activities were estimated with viper 1.22.0[109] for each cell or pseudobulked subcluster, based on the regulons from DoRothEA v1.3.0[107,110]. Pseudobulking was performed using the AverageExpression function from Seurat with default parameters. The scores were summarized per subcluster and most variable TFs selected for plotting.

**Processing of public single-cell data**. Publicly available datasets from the studies by refs.[23,15] were downloaded from the respective resource. Datasets were processed as close as possible to the description in the methods section of the corresponding publication, including data filtering and cluster annotation. We noticed minor differences in the resulting clusters e.g., McLellan describes 9 fibroblast subclusters, while we obtained 11 subclusters. We suspect differences in the cell ranger version and R packages to be responsible for the difference, however the clusters relevant for the comparison were re-identified in the public data.

**Score-gene correlation**. The general strategy for the correlation analysis was adopted from ref.[111] and extended from gene-gene to score-gene correlation. Therefore, instead of correlating a single gene to all other genes, we correlated the ECM score (see section function scoring) to all genes, in order to identify correlating genes. Genes included in the ECM gene set needed to be excluded from the correlation, since they provide the basis for the score and would be found as highest correlating. Similar to the approach described by Mayr et al. we performed modularity optimization using the Louvain algorithm with the FindClusters function from Seurat at a relatively high-resolution parameter (set to 10). This generated a large number of transcriptionally similar cell clusters and averaging gene expression as well as ECM scores across these meta-cells mitigated the impact of sparse counts at the single-cell level and increased correlation values. Pearson correlation with the averages of the meta-cells was used to detect non-ECM genes correlating in expression to the ECM score. Enriched gene sets within the highest correlating genes were analyzed as described in the GSEA section.

**Score-score correlation**. To identify TF activities correlating to an increase in ECM gene expression, we tested for predicted TF activity scores correlating with the ECM score. Therefore, we estimated TF activities as described above with viper and DoRothEA. Average activity scores per meta-cell were calculated and tested by Pearson correlation against the ECM score. The top 10 correlating TF activity scores were visualized as a heatmap, sorted by ECM score.

**Cluster correlation**. To compare fibroblast subclusters from other studies of murine cardiac diseases models, we downloaded and processed the corresponding publicly available datasets (see also section "Processing of public single-cell data"). In order to perform correlation analysis of clusters, we merged the fibroblast data from our study with the processed and annotated public datasets. Next, we calculated average gene expressions per fibroblast subcluster and obtained a gene matrix of the 500 most variable genes of the merged dataset (calculated by FindVariableFeature, with default). This matrix was used for Pearson correlation analysis and the resulting coefficient values were plotted as a heatmap.

**RNA velocity and trajectory analysis**. The command line function from velocyto[112] was used to generate spliced/unspliced expression matrices for each fibroblast sample based on the cell ranger output. Matrices were loaded into R with the SeuratWrapper function ReadVelocity. Each sample was filtered for fibroblasts that remained in the Seurat integration. Furthermore, genes with a high potential to be stress induced by the cell isolation method were excluded from the matrices. Matrices were merged per condition (sham, TAC 14 days, TAC 28 days) and UMAP embedding of the Seurat integration was added to the three resulting samples. Next, RNA velocity analysis was performed by scVelo[113] separately for the fibroblasts per condition. scVelo was executed with default parameters and dynamic modeling was used to estimate velocities. Latent time was also calculated with default parameters.

RNA trajectory analysis was performed by monocle 3[114–116]. A trajectory was selected from Fib1 to ECM-Fib and pseudotime calculated starting at Fib1. DEG across the trajectory were calculated with the graph_test function separately per condition. The top 500 significant genes of each test were combined and clustered as described in the DEG section for heatmap visualization. Enriched gene sets per gene cluster were analyzed as described in the GSEA section. Smooth functions for the gene expression measures along pseudotime were plotted using the geom_smooth function (ggplot2 package) with method set to generalized additive model, default formula and distribution set to negative binomial.

**Ligand receptor mediated cell-cell communication**. The LR analysis was performed using the CellPhoneDB (CPDB, Version 2.0.5[117]). Initially, scRNA-seq matrices were log-normalized and scaled using Seurat functions. The murine gene names were assigned to respective human orthologous (HUGO annotation) using biomaRt (Version 1.2.0[118,119]) and EWCE (Version 0.99.2[120]). Expression matrices were splitted according to respective condition (i.e., TAC and sham). Then, CPDB was executed for each dataset using the "statistical_analysis" method. Aiming to increase the reliability of LR inference CPDB was fed with a database enhanced by the combination of five different LR data sources (CPDB[117]; TalkLR[121]; scTensor[122]; SCA[123]; iTALK[124]), interactions that presented at least two consensus data sources were kept in the final LR database. Using the statistically significant interactions ($p$ value < 0.05) from CPDB output, ranking and the visualization were generated by CrossTalkeR[49] To perform the annotation of the LR interactions, the ligands/receptors set was splitted in the following two subsets (1) upregulated (i.e., MeanLR > 0) and (2) downregulated (i.e., MeanLR < 0). These subsets contain exclusive ligands and receptors; the genes which at the intersection of this were disregarded. The process results in four distinct gene sets (1) Ligand upregulated, (2) Ligand downregulated, (3) Receptor upregulated and (4) Receptor downregulated, each gene set was annotated to pathways databases by using the packages ClusterProfiler[125] and ReactomePA[126].

**Statistics and reproducibility**. Unless otherwise stated, statistical significance was assessed by a two-tailed Student's $t$ test or one-way ANOVA with Tukey's or Dunnett's multiple comparison with a $p$ value < 0.05 being considered statistically significant. Statistical analyses were performed using GraphPad Prism 9.0.1. GSEA is performed with the hypergeometric test for overrepresentation (one-sided), followed by multiple comparison correction. The individual method of correction for gProlifer2 (algorithm g:SCS[127]) and enrichR[103] is described in the corresponding publications. All stainings were performed in triplicates.

**Reporting summary**. Further information on research design is available in the Nature Research Reporting Summary linked to this article.

## Data availability

The single-cell RNA sequencing data generated in this study have been deposited in the Gene Expression Omnibus database under accession code GSE166403. All other relevant data supporting the key findings of this study are available within the article and its Supplementary Information files or from the corresponding author upon reasonable request. Source data are provided with this paper.

## Code availability

Custom scripts used in single-cell data analysis are available at: https://github.com/KramannLab/Murine_heart_map.

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

## Acknowledgements
This work was supported by grants of the German Research Foundation (DFG: SFBTRR219 322900939, CRU344- 4288578857858, CRU5011- 445703531) by a Grant of the European Research Council (ERC-StG 677448), a Grant of the Else Kroener Fresenius Foundation (EKFS), the Dutch Kidney Foundation (DKF), TASKFORCE EP1805 and by the ERA-CVD MENDAGE consortium (BMBF 01KL1907), the NWO VIDI 09150172010072 and a Grant from the Leducq Foundation all to R.K. Research from M.H. is in part funded via the European Union's Horizon 2020 research and innovation programme under the Marie Skłodowska-Curie grant agreement No 722609. This work was also supported by the BMBF eMed Consortia Fibromap (to R.K., R.K.S. and I.C.) and by the German Research Foundation (CRU344, Z) to I.C.

## Author contributions
R.K. designed the study and F.P., M.H. and R.K. interpreted the data. F.P., M.H., I.C. and R.K. designed the data analysis plan. F.P. and M.H. contributed equally to writing the manuscript and organizing the Figs. I.C., R.K.S., K.H., M.L., C.K. and M.T.S. edited the manuscript and advised on data interpretation. F.P. carried out most single-cell data analysis. J.N. carried out cell-cell communication analysis. F.P. and R.L. carried out the RNA velocity and trajectory analysis. J.M. performed all mouse surgeries. F.P. and M.H. carried out all single-cell and imaging experiments. N.K. and C.S. provided tissue from human HF patients. S.Z. designed and cloned plasmids. E.M.J.B. sequenced the single-cell libraries. F.P., M.H. and R.K. initiated the study. All authors read and approved the final manuscript.

## Competing interests
The authors declare no competing interests.
