## [Peer Review File · Nature Communications]

REVIEWER COMMENTS

Reviewer #1 (Remarks to the Author):

This interesting study assesses the cell composition and expressional phenotype in genetic murine models that were submitted to TAC. Using an elegant bioinformatic approach the authors recognises some interesting deviations that could be instrumental to the fibrotic reaction and disconnection of the link between angiogenesis and repair.

The study has however some important limitations especially in the functional characterization of the phenotype.

1. The model of TAC is an acute inflammatory situation used frequently in the lab but very distant from chronic heart failure. This diminishes the translational value of the data.
2. The analysis is performed in the apex, but no justification is given or data shown the changes can be generalized.
3. The use of pseudo-time is not sufficient to dissect the temporal changes occurring in the different cell subsets. This is a single picture, but cells could rapidly modify their phenotype during the evolution of the remodelling. Therefore, interpretation should be very cautious.
4. The study does not have any hemodynamic or functional assessment to integrate the molecular-histological data.
5. There is no attempt to rescue the cell phenotype even at in vitro level, to understand if the fibroblast or pericyte subpopulation could be reprogrammed to be less profibrotic by modulating one of the discovered hit/target.
6. Some statements, such the disconnection between fibroblasts and vascular cells, are only based on molecular endpoints. No experiment was done using coculture of the relevant subpopulations. How the authors can exclude that the molecular landscape of endothelial cells is due to an altered mechano-sensing or interaction with cardiomyocytes?

Reviewer #2 (Remarks to the Author):

This is a well-executed descriptive study using fate mapping approaches and single cell RNA-seq to define the heterogeneity of endothelial cells, mural cells and fibroblasts in the pressure-overloaded myocardium. The study is systematic, the data are of outstanding quality. Interpretation and discussion are generally sound. The main weakness is the purely descriptive nature of the study, and the reliance on transcriptomic analysis to derive conclusions on functional phenotypes. Considering the large amount of high-quality single cell transcriptomic data provided, this is a relatively minor detractor. The following concerns, need to be addressed:

Major comments:

1. Some generalized conclusions are not adequately supported by data. For example, the authors state that pericytes "play a role in cardiac fibrosis and remodeling". In the absence of functional data, this conclusion should be toned down, as the transcriptomic data provided by the authors do not document such a role. Moreover, the authors need to better define how the observed transcriptomic profiles of mural cell subsets correspond to specific functional phenotypes.
2. It is not clear whether the subpopulations of fibroblasts, endothelial and mural cells noted following TAC were present in normal hearts (with very similar profiles) and simply expanded following stress, or if there were certain subsets that appeared de novo in the pressure-overloaded myocardium. The reader gets the impression that the former model is correct. Does this simplified statement reflect the authors' data?
3. The authors describe 7 subsets of fibroblasts on the basis of single cell transcriptomic analysis. Although the reviewer recognizes the challenges in defining functional properties

of various subsets on the basis of transcriptomic profiles, a better definition (or at least clarification of the definitions provided) may be possible. Fib1 is suggested to have higher collagen expression. However, only collagen IV is indicated as upregulated in this subset. Do these cells produce only basement membrane collagens? How about structural collagens I and III? Fib2 are labeled as "stressed fibroblasts". How is this justified? Were these simply dying cells (expressing apoptosis-related genes)? "Matrix fibroblasts" seem to be periostin-high and also express other matricellular genes. Are these myofibroblasts? Do they express Acta2? Do they also express fibrillar collagens? How do the subpopulations identified in the current study relate to previously identified subsets using traditional approaches (for example α -SMA+ or Postn+ myofibroblasts, FAP+ activated fibroblasts, etc) or using scRNA-seq.

4. Considering the role of proteases/antiproteases in cardiac remodeling and the potential for distinct roles of "matrix-degrading" vs "matrix-preserving" cells, it would be informative to ask whether some of the subpopulations identified by the authors exhibit distinct expression patterns of MMPs/TIMPs.

5. Regarding endothelial cell phenotype: the authors point out that "further analysis pointed towards failed angiogenesis". How is this conclusion supported by the data? Did the authors observe capillary rarefaction?

6. A limitation of the study is the investigation of a single timepoint (2 weeks after TAC). Pressure overload results in a sequence of cellular and functional changes that cannot be recapitulated through studies of a single timepoint. The reviewer recognizes that the type of analysis performed here cannot be easily repeated at multiple timepoints. However, the authors need to discuss the limitations.

7. The authors state that the PDGFRbetaCre-derived population expanded following TAC; these cells exhibited fibroblast phenotype. Does this driver label both fibroblasts and mural cells? Does it label all fibroblasts?

8. The abstract requires revisions and clarifications: a) "We identified that injury causes a loss of cellular cross-talk towards endothelium in the vascular niche". The meaning of this sentence is unclear. Please revise and be more specific. b) The final sentence of the abstract the authors state that a "previously unappreciated role of pericytes in cardiac remodeling and fibrosis was revealed". However, the rest of the abstract does not include description of data supporting this conclusion. Please add a couple of sentences that support this final conclusion.

Reviewer #3 (Remarks to the Author):

The manuscript by Peisker and Kramann and colleagues probes non-cardiomyocyte cell heterogeneity in a murine model of heart failure due to pressure overload (transaortic constriction; TAC) using single cell transcriptomics. This work was facilitated by cell subpopulation enrichment using Cre-based cell marking in sham and TAC conditions using a variety of Cre strains generally specific for fibroblasts, endothelial cells and mural cells. Cre strains were also used to highlight and clarify the spatial distribution of marked cells and the apparently rare deviations from baseline cell fates using immunostaining. The choice of the TAC model and 2-week timepoint allows for the development of pressure-overload heart failure that may resemble the impact of hypertension in humans, adding to the myocardial infarction and AngII-induced hypertension single cell datasets already published, albeit that hypertension in humans would be a chronic disease developing over years if not decades. Thus, whereas TAC is a commonly-used model, its relationship to human high blood pressure is debatable. Furthermore, results may be expected to be similar to the AngII model.

In general, this is a descriptive paper that would nonetheless add significant new data to previous sc/snRNAseq analyses of cardiac homeostasis and disease.

Major Issues:

1. A strength of the paper is the enrichment for interstitial cells subtypes before scRNAseq; however, this comes at the cost of having only a single time point, thus the power to resolve earlier cell states and cell trajectories is markedly diminished.
2. The authors embark on a detailed lineage marking exercise that allows cell purification (with retrospective identification of marked cells by scRNAseq) and tracing of lineage descendants; however, there is no clear aim or model provided for this section. This is a missed opportunity. Given that much has been made of the potential plasticity of stromal cells, clear goals and outcomes (and limitations) would strengthen this paper.
3. In Figure 1C, it is apparent that not all Cre crosses showed significant hypertrophic changes – specifically *Pdgfrb*-Cre and *Myh11*-Cre. In others, significance seems to be driven by a few examples. Given the 2-week time point when TAC hearts should be showing hypertrophy (increased heart weight to tibial length), this suggests technical variation in the TAC surgery, which could add considerable noise to the data and raising concerns about analysis of population proportions. The authors should have checked the pressures on either side of the constriction.
4. For the statistical testing of differences in population proportions between sham and TAC (e.g. Fig 1g), have the authors tried a differential proportion test developed for single-cell data, e.g. scDC (Cao et al. BMC Bioinformatics 2019) or DPA (Farbehi et al., eLife 2019). A standard contingency table test will not be able to account for the inherent technical variability in scRNA-seq data.
5. There are a high number of samples processed, requiring batch correction. It appears that the challenges of data integration were only considered retrospectively: “..it became apparent to us that the integration of all 12 data sets requires a different strategy than integrating two or six datasets.”.
6. One of the puzzling outcomes of this paper is the high similarity between sham and TAC samples. Only one population, MatriF, expresses Periostin, a marker of activated fibroblasts and myofibroblasts. This is also the only population to express *Thbs4*. It seems incongruous that these are present in abundance in Sham samples and increase to a fairly modest degree after TAC and I suspect that this may be due to technical challenges of batch correction. The authors use Seurat and Harmony batch correction methods to combine their datasets. Given that they are correcting between sham and TAC, there may be a risk for ‘over-correction’ such that cell states that should be specific to TAC might end up observed in sham or vice versa, thus confusing the biological interpretation of the data. Indeed, as noted, among fibroblasts about 40% of the MatriF cells are in sham condition, which seems improbable given findings from other scRNA-seq studies comparing fibroblasts in sham and diseased hearts (e.g. MacLellan et al., Circulation 2020, Forte et al. Cell Reports 2020, Farbehi et al., eLife 2019). In MacLellan et al., which may be the most relevant to this study, their *Thbs4*-fibroblast population (which may be similar to the author’s MatriF cells) is essentially unique to their diseased condition. The authors should consider the impact of batch correction on interpretation and compare to results obtained from not correcting between sham and TAC (e.g. analysing sham and TAC separately).
7. The use of the term MatriF to describe the Periostin+/*Thbs4*+ fibroblast subset is confusing given its similarity to matrifibrocytes, which I do not think features in this model, but are a prominent sub-state in the MI model. Nomenclature is going to be tricky in the single cell field without unintentional distractions; thus I suggest choosing an alternate name.
8. The authors report numerous stressed cell types in their lineage-traced datasets. Consistently, these populations appear to be represented more in sham than TAC, indicating that the stressed states are likely to be due to technical rather than biological factors. In this case, the stressed cells should be removed from the data prior to other analysis such as differential expression.

9. Pertinent to points 7 and 8, the authors neglect to compare their data with previous cardiac scRNA-seq studies, which includes one study on an AngII hypertension model. Again, this is a missed opportunity and further complicates the field with an additional boutique nomenclature. As noted, there is the potential for technical artifacts in batch-correction that skew cell states - obviously the authors should address this possibility first. 9. The authors raise a question in Discussion about the potential for MSC-like cells among fibroblasts. This is an issue that has been discussed previously in scRNA-seq studies focussed on cardiac fibroblasts, with MSC or progenitor-like fibroblast sub-populations reported (Forte et al. Cell Reports 2020, Farbehi et al., eLife 2019; Solimon et al., Cell Stem Cell 2020). The authors should include these previous findings in their discussion of MSC-like fibroblasts and in general better contextualise their work in light of previous papers in this field.

Other comments:

- 1. There are numerous spelling and grammatical errors throughout the manuscript, please check carefully.**
- 2. Reference to the model in McLellan et al. 2020 as myocardial infarction should be corrected.**

The authors would like to thank all the reviewers for their expertise and careful assessment of our paper. We have taken all the comments into account and present to you a substantially modified and further improved manuscript, providing extensive new data in response to your comments.

In response to the reviewers' comments, we have now included the following salient additional data:

Single cell RNA sequencing data from a second time point 28 days after TAC for fibroblast and endothelial cells, provide insight into transcriptional changes occurring later in progressive hypertrophic remodeling.

Validation of fibroblast heterogeneity by comparison to the publicly available datasets generated for the studies of Forte et al. (Cell Reports 2020 PMID: 32130914) and McLellan et al. (Circulation 2020 PMID 32795101).

Multiple RNA *in situ* hybridisation stainings including: *Col1a1* on Myh11CreER fate traced samples labeling mural cells; *Postn* of sham hearts; *Atf3*, *Ifit1*, *Wif1*, *Fgl2*, *Pdgfra* on multiple murine heart tissue slides, *PDGFRa* and *THBS4* on a human heart tissue from heart failure patient

In vitro validation of transcription factors with predicted activation after TAC, revealing TEAD1 as a pro-fibrotic transcriptional regulator.

Quantification data of endothelial cell proliferation by image quantification.

Furthermore we have largely rewritten the results and discussion to integrate our new data and provide clear statements on the limitations of our study.

Reviewer #1 (Remarks to the Author):

This interesting study assesses the cell composition and expressional phenotype in genetic murine models that were submitted to TAC. Using an elegant bioinformatic approach the authors recognises some interesting deviations that could be instrumental to the fibrotic reaction and disconnection of the link between angiogenesis and repair.

We thank the reviewer for the overall very positive evaluation of our work.

The study has however some important limitations especially in the functional characterization of the phenotype.

1. The model of TAC is an acute inflammatory situation used frequently in the lab but very distant from chronic heart failure. This diminishes the translational value of the data.

We appreciate the concerns raised by the reviewer and understand the limitations of the model with regards to clinical relevance. However, it has been demonstrated that certain findings of this model can indeed be translated. We would like to point out that *Thbs4*, described in our study as a highly specific marker for injury related fibroblasts, was already identified in multiple studies of human heart failure (Tan et al. Proc Natl Acad Sci USA 2002 PMID: 12177426; Argenziano et al. PLoS One 2019 PMID: 31083689; Rao et al. Basic Res Cardiol 2021 PMID: 34601654; Koenig et al. bioRxiv 2021 DOI: 10.1101/2021.07.06.451312; Kuppe et al. bioRxiv 2020

DOI: 10.1101/2020.12.08.411686) including a single cell RNA sequencing based study by Rao et al., which identified a *Postn*⁺/*Thbs4*⁺ fibroblast cluster from diseased heart samples. Experimental studies in murine *Thbs4* knockout models revealed its importance for cardiac fibrosis and remodeling (Frolova et al. FASEB 2012 PMID: 22362893). We have now also confirmed that *Thbs4* was present in the human MI heart and colocalizes with *Pdgfra*⁺ fibroblasts by RNA *in situ* hybridization (new Extended Data Figure 5c). Furthermore, a recent publication in Nature (Alexanian et al. Nature 2021 PMID: 34163071) used data generated on TAC mouse models to identify MEOX1 as an important transcription factor for fibroblast activation, which was also found in our analysis (new Figure 4a). Alexanian et al. then demonstrated that MEOX1 is also a critical driver of human fibroblast activation.

We agree that our time point of 14 days after surgery might reflect rather acute remodeling processes compared to an established chronic phase. We therefore added a later time point of 28 days after surgery for our fibroblast and endothelial cell data. A systematic study of the TAC model by Platt and colleagues showed that within the first 4 weeks after the surgery most of the fibrotic remodeling occurs, together with strong hemodynamic changes (Platt et al. Front Physiol 2018 PMID: 29867532; Richards et al. Sci Rep 2019 PMID: 30971724). However, after this earlier remodeling the function slowly declines to a decompensated state, which was not captured by our study. We added this limitation to our discussion. Regarding insights from data of the new time point 28 days after TAC, we would like to highlight the following findings. For fibroblast, we found a significant continuous expansion of ECM-Fib, which was most prominent at the latest time point (new Fig. 3f and new Extended Data Fig. 4d). Furthermore we predicted increasing TEAD1 transcription factor activity and validated a pro-fibrotic role for TEAD1 *in vitro* (new Fig. 4). For endothelial cells we found reduced proliferation towards 28 days after TAC (new Fig. 6g, h and new Extended Data Fig. 8e) and detected a larger amount of differentially expressed genes associated with angiogenesis, hypoxia signaling and cell migration (new Fig. 6k, l and new Extended Data Fig. 9b-e). Taken together, we improved our coverage of remodeling processes of the vascular niche towards later hypertrophic remodeling.

2. The analysis is performed in the apex, but no justification is given or data shown the changes can be generalized.

We apologize for this misunderstanding which is likely due to the phrasing in our methods. We have initially taken the entire apical half of the mouse heart not only the apex. We have then prepped the left ventricle from this part of the heart and have utilized the other half of the mouse heart for imaging studies. Therefore, the LV part that we have used for scRNA-sequencing contains the entire apex but also septum, lateral, anterior and posterior left-ventricular wall and therefore represents large parts of the left ventricle. We solely lack the basal parts of the ventricle with the valves. We apologize for not making this clear and have changed the text now accordingly.

"We used the apical half of the left ventricle (including apex, anterior-, lateral-, posterior-wall and septum) to generate a single cell suspension with subsequent FACS enrichment for tdTom⁺, viable (DAPI) cells for scRNA-seq (10x Genomics). The basal part of the left ventricle was used for confocal imaging analysis (Fig. 1e)."

3. The use of pseudo-time is not sufficient to dissect the temporal changes occurring in the different cell subsets. This is a single picture, but cells could rapidly modify their phenotype during the evolution of the remodeling. Therefore, interpretation should be very cautious.

We agree with the concerns raised regarding pseudotime based analysis of cell fates. Trajectory and pseudotime analysis can only provide an artificial model of differentiation processes. Therefore we have now added a second time point 28 days after TAC surgery to be able to look into changes that occurred at a later time point and integrated this later time point with our trajectory analysis (new Fig. 5a-f and new Extended Data Fig. 7c-g). Results from our new RNA velocity analysis provided further evidence for Fib1 as origin for ECM-Fib (new Fig. 5a, b and new Extended Data Fig. 7c, d). We used Monocle 3 to find trajectories in our fibroblast data and focused on a predicted path from Fib1 to ECM-Fib (new Fig. 5c and new Extended Data Fig. 7e). Analysis of the distribution of fibroblasts over pseudotime (separate for each condition) shows an increased enrichment of fibroblasts towards the end of the pseudotime when comparing sham, TAC 14 days and TAC 28 days (new Fig. 5d). This reflects the observed continuous expansion of ECM-Fib after TAC (new Fig. 3f). We think our data suggests a differentiation path from Fib1 to ECM-Fib after TAC induced injury, however, we also need to point out that further studies will need to validate this computational model by more complex in vivo lineage tracing approaches. We have indicated potential caveats of this analysis in the manuscript.

"Our trajectory interference analysis suggested a more general pool of fibroblasts as potential progenitors, mostly consisting of the Fib1 type, however, this computational model needs to be considered carefully. The progenitor role of cardiac fibroblasts remains controversial and more experiments are needed to dissect the progenitor potential of the different cardiac subtypes."

4. The study does not have any hemodynamic or functional assessment to integrate the molecular-histological data.

We agree that this would be a nice addition. Unfortunately, we have not performed functional or hemodynamic assessment on the day 14 TAC experiments. However, since we now added a later time point for *Gli1CreER* and *Cdh5CreER* mice, we performed echocardiography in these mice at 14 and 28 days after TAC surgery to measure the peak pressure across the stenosis and the left ventricular function (new Extended Data Fig. 1a). This analysis demonstrated significantly increased pressure trans the TAC- stenosis and also reduced left ventricular ejection fraction in both mouse lines.

5. There is no attempt to rescue the cell phenotype even at in vitro level, to understand if the fibroblast or pericyte subpopulation could be reprogrammed to be less profibrotic by modulating one of the discovered hit/target.

We agree and appreciate this suggestion. Our aim was to map vascular and perivascular cells in a mouse model of heart failure to describe and identify mechanisms of cardiac remodeling. We have validated the role of several genes in

myofibroblast differentiation *in vitro*. Our scRNA-seq analysis indicated an increased activity for the transcription factor TEAD1 in pro-fibrotic fibroblast differentiation suggesting TEAD1 as being potentially involved in this process. We now provide novel data using retroviral TEAD1 overexpression in murine cardiac Gli1⁺ cells myofibroblast differentiation assays (new Fig. 4i-j and new Extended Data Fig. 5g). These experiments demonstrated that TEAD1 was able to induce myofibroblast differentiation even in the absence of TGFβ with increased *Acta2*, *Col1a1* and fibronectin expression (new Data Fig. 4j). Furthermore, overexpression of TEAD1 amplified the response of Gli1⁺ cells to TGFβ and further increased expression of *Acta2* and fibronectin as compared to cells with expression of a control virus (new Fig. 4j and new Extended Data Fig. 5g). This data indicates that TEAD1 is a novel regulator of fibroblast activation. We performed similar experiments with retroviral overexpression for Klf5 and Pax6, since we found a high activity score in fibroblasts for these TF after 14 days TAC (Rebuttal Fig. 1a). However, we did not observe an effect on collagen 1 or *Acta2* expression (Rebuttal Fig. 1b) and furthermore, we did not observe a higher score for Klf5 and Pax6 at our new later time point of TAC 28d (new Fig. 4f, g).

Rebuttal Fig. 1: Validation of potentially profibrotic transcription factors

a, Transcription factor activity prediction based on the TAC 14 d time point. *Pax6*, *Klf5* and *Tead1* were selected for *in vitro* validation. Results for *Tead1* are shown in the new Extended Data Fig. 5g. **b**, qPCR quantification of *Acta2* and *Col1a1* expression after retroviral *Klf5* and *Pax6* overexpression, combined with TGFβ stimulation.

6. Some statements, such the disconnection between fibroblasts and vascular cells, are only based on molecular endpoints. No experiment was done using coculture of the relevant subpopulations. How the authors can exclude that the molecular landscape of endothelial cells is due to an altered mechano-sensing or interaction with cardiomyocytes?

We agree with the reviewer that this is a limitation of the study. Unfortunately our experimental design of lineage tracing combined with enrichment for tdTom⁺ cells excluded cardiomyocytes. Therefore, we are not able to perform ligand receptor analysis of cardiomyocytes and cells of the vascular niche. We also did not perform actual co-culture experiments in this regard. We added this as a limitation to our discussion part. We however can speculate on the influence of increased ECM deposition due to hypertrophic remodeling on the mechanosensing ability of endothelial cells based on mRNA expression. Ligand receptor analysis shows that

focal adhesion signaling is increased towards artery EC (new Fig. 6m). This suggests that mechanosensing might affect this EC subtype.

We now added the following text to the results and discussion sections:

"Changes in cellular interaction were highly related to ECM-Fib and ArtEC including PI3K-AKT signaling and focal adhesion complex interactions. These functional complexes likely play a central role in the niche interaction changes, since the ECM composition is changed by the ECM-Fib, which in turn can be sensed by the ArtEC. Interestingly, targeting integrin subtypes, which are the connection between the focal adhesion complex and the ECM, has already been shown as promising therapeutic targets in fibrotic remodeling (Henderson et al. Nat Med 2013 PMID: 24216753) ."

"Additionally, we acknowledge that our lineage tracing based enrichment strategy excluded cardiomyocytes, therefore we lack analysis of the potential influence from this cell population."

Reviewer #2 (Remarks to the Author):

This is a well-executed descriptive study using fate mapping approaches and single cell RNA-seq to define the heterogeneity of endothelial cells, mural cells and fibroblasts in the pressure-overloaded myocardium. The study is systematic, the data are of outstanding quality. Interpretation and discussion are generally sound. The main weakness is the purely descriptive nature of the study, and the reliance on transcriptomic analysis to derive conclusions on functional phenotypes. Considering the large amount of high-quality single cell transcriptomic data provided, this is a relatively minor detractor. The following concerns, need to be addressed:

We appreciate this comment and thank the reviewer for the overall positive assessment of our work.

Major comments:

1. Some generalized conclusions are not adequately supported by data. For example, the authors state that pericytes "play a role in cardiac fibrosis and remodeling". In the absence of functional data, this conclusion should be toned down, as the transcriptomic data provided by the authors do not document such a role. Moreover, the authors need to better define how the observed transcriptomic profiles of mural cell subsets correspond to specific functional phenotypes.

We agree with the reviewer that the data does not sufficiently support the statement about the role of pericytes in cardiac fibrosis. We now performed differential gene expression analyses in Peri1 and Peri2 demonstrating increased expression of different collagen subtypes in the TAC condition (new Extended Data Fig. 11b). In addition, we filtered our functional association data to further define differences between different mural cell types (new Extended Data Fig. 10e). We further performed RNA *in situ* hybridization experiments with different strategies to specifically stain pericytes and were able to detect a few *Col1a1* expressing pericytes in murine hearts to verify our scRNA-seq data (new Extended Data Fig. 11c). However, we agree that this is not a substantial contribution and that functional data is needed to verify the contribution of pericytes to cardiac remodeling. We therefore toned down our conclusion about this data and the

contribution of pericytes to cardiac fibrosis in the abstract, results and discussion part.

2. It is not clear whether the subpopulations of fibroblasts, endothelial and mural cells noted following TAC were present in normal hearts (with very similar profiles) and simply expanded following stress, or if there were certain subsets that appeared de novo in the pressure-overloaded myocardium. The reader gets the impression that the former model is correct. Does this simplified statement reflect the authors' data?

We thank the reviewer for this interesting question. To give a clear answer to this question, it is very important to define what a subset is. Transcriptional changes occur in all fibroblasts after TAC, but it remains difficult to draw the line between a change in the subtype state or true differentiation towards a different subtype. Our data indeed suggests that the mentioned cells are already present in non-injured hearts. One example are the ECM-Fib that are defined by high *Postn* expression. We have identified this population in our sham datasets and also demonstrated that this population of *Postn*⁺ cells is present in non-injured murine hearts from other published datasets including Forte et al. (Cell Reports 2020 PMID: 32130914) and McLellan et al. (Circulation 2020 PMID 32795101) (new Extended Data Fig. 7a and Rebuttal Fig. 2). Transferring the cluster labels from the complete annotation of the corresponding data set and comparison to the expression pattern of *Postn* clearly shows a population of fibroblasts, which resembles ECM-Fib (MFC for Forte et al., FibThbs4/FibCilp for McLellan) (new Extended Data Fig. 7a and Rebuttal Fig. 2). Furthermore, we also performed RNA in situ hybridisation for *Postn* in non-injured (sham) murine hearts and demonstrated the presence of *tdtom*⁺ *Postn* expressing cells in homeostasis (new Extended Data Fig. 7b).

We have now added the following section to our results part:

"We identified ECM-Fib as fibroblast subtype with the highest matrix related gene expression and strongest expansion after TAC (Fig. 3f). Interestingly, our data suggested that a small fraction of these cells were already present in homeostasis prior to injury (sham) (Fig. 3f and Extended Fig. 4d). To exclude that this finding was an integration artifact, we separately clustered fibroblasts from only our sham samples as well as fibroblasts from the control data from Forte et al. (Cell Reports 2020 PMID: 32130914) and McLellan et al. (Circulation 2020 PMID 32795101) (Extended Data Fig. 7a). Transferring the cluster labels from the complete annotation and comparison to the expression pattern of Postn clearly shows a population of fibroblasts in the control datasets, which resembles ECM-Fib (Extended Data Fig. 7a). In addition, in situ hybridization experiments identified Postn expressing fibroblasts (Pdgfra⁺) in sham hearts (Extended Data Fig. 7b). This provides strong evidence of a, albeit minor, non-activated ECM-Fib population present in homeostasis that lacks Thbs4 expression."

Rebuttal Fig. 2: ECM-Fib / Periostin high fibroblast in control datasets

a-c Analysis of control heart derived fibroblast. (a) sham fibroblast from this study, (b) control fibroblast from McLellan et al., (c) control fibroblast from Forte et al.. UMAP on the left shows the full fibroblast dataset. Fibroblast from the control datasets were subsetted and clustered independent of the disease samples, generating a new UMAP (3 central UMAP plots). *Postn* expressing cells are clustering together (color of the UMAP indicates level of expression). Two violin plots summarize the *Postn* expression per cluster, upper panel for the unsupervised clustering of control fibroblast, lower panel for the transferred cluster labels from the full annotation.

3. The authors describe 7 subsets of fibroblasts on the basis of single cell transcriptomic analysis. Although the reviewer recognizes the challenges in defining functional properties of various subsets on the basis of transcriptomic profiles, a better definition (or at least clarification of the definitions provided) may be possible. Fib1 is suggested to have higher collagen expression. However, only collagen IV is indicated as upregulated in this subset. Do these cells produce only

basement membrane collagens? How about structural collagens I and III? Fib2 are labeled as “stressed fibroblasts”. How is this justified? Were these simply dying cells (expressing apoptosis-related genes)? “Matrix fibroblasts” seem to be periostin-high and also express other matricellular genes. Are these myofibroblasts? Do they express Acta2? Do they also express fibrillar collagens? How do the subpopulations identified in the current study relate to previously identified subsets using traditional approaches (for example α -SMA+ or Postn+ myofibroblasts, FAP+ activated fibroblasts, etc) or using scRNA-seq.

We thank the reviewer for raising these important questions regarding fibroblast heterogeneity and functional differences of fibroblast subtypes. In the following, we provide a better definition of the fibroblast subsets identified in this study. Fib1 were associated with the second highest ECM score and showed a strong increase in the expression of network forming collagens 14 days after TAC (new Fig. 3g, h). Fibrillar collagens were also increased to this time point, but the expression decreased towards 28 days after TAC. The increase of network forming collagen expression was mostly related to Col4 and Col6 in Fib1 (new Extended Data Fig. 4h). Col4 is a major component of basement membrane and Col6 interacts with the basement membrane, indicating a strong role in basement membrane remodeling for Fib1. ECM-Fib (renamed from “Matrix fibroblasts”) showed a strong expression of fibrillar collagens 14 days after TAC, which was lower at 28 days, but still higher compared to sham (new Fig. 3h). Overall ECM-Fib were associated with the highest increase in ECM related gene expression (new Fig. 3g). Therefore the function of ECM-Fib seems to be more associated with structural support of the surrounding tissue.

Regarding the fibroblast subtype previously named “stressed fibroblasts” (now named “Atf3-Fib”), we now performed RNA *in situ* hybridization and comparison to publicly available datasets. We found *Atf3* as highest expressed marker gene of the subtype (new Fig. 3d) and we were able to detect *Atf3* expressing fibroblasts in cardiac tissue by RNA *in situ* hybridization (new Extended Data Fig. 4b). The comparison of our data with the public data sets from Forte et al. (Cell Reports 2020 PMID: 32130914) and McLellan et al. (Circulation 2020 PMID 32795101) provides further evidence for the fibroblast heterogeneity of the murine heart. These two publications performed single cell RNA sequencing of the heart after myocardial infarction (MI, Forte et al.) and angiotensin 2 (AngII, McLellan) induced remodeling. Similar to our dataset, *Atf3* expression occurred in a clustered pattern in both public datasets (Rebuttal Fig. 3 and new Extended Data Fig. 6f, g), even though a specific cluster of *Atf3* expressing fibroblasts was not described there. Furthermore, a recent publication by Forsström et al. (Cell metabolism 2019 PMID: 31523008) described *Atf3* in myoblasts to be activated towards the terminal stage of mitochondrial integrated stress response. The integrated stress response is a complex cellular program, which can get activated by different conditions and result in cell survival or cell death, however *Atf3* is reported to be involved in the cellular survival response (Pakos-Zebrucka et al. EMBO Rep. 2016 PMID: 27629041). Here we would like to point out that filtering for apoptotic or in general dying cells in single cell RNA sequencing data is a standard procedure. Usually the percentual mitochondrial RNA (%-mt) content in those cells is higher than in the general population (Luecken and Theis Mol Syst Biol 2019 PMID: 31217225). We used a stringent cut off value of >6% to include only cells with a low %-mt content and exclude dying and low quality cells. Taken together, we provide evidence that the *Atf3*-Fib subtype is part of the true biological heterogeneity of cardiac

fibroblasts. However, we cannot exclude that *Atf3*-Fib were more susceptible to transcriptomic changes by the isolation methods used by us and others. Therefore we did not perform more specific analysis on these subtypes.

Regarding the further question on cardiac fibroblast subtypes, we compared our fibroblast subtypes with subtypes defined in the respective studies by correlation analysis (new Extended Data Fig. 6a). This led to the identification of at least three conserved fibroblast subtypes, which seem to be covered by all three datasets. (1) Fib2 in our study were highly similar to Fib5 of the McLellan study and progenitor-like state (PLS) fibroblast characterized in the study by Forte. Conserved marker genes of this cluster are *Cd248*, *Ly6a* and *Pi16*. There are multiple recent studies suggesting fibroblast with *Ly6a* expression as progenitor population (Tang et al. *Circulation* 2018 PMID: 30566021; Soliman et al. *Cell Stem Cell* 2020 PMID: 31978365; Buechler et al. *Nature* 2021 PMID: 33981032). We were not able to find any transcriptomic feature for this population indicating a specific progenitor function, however a more focused experimental design will be necessary to clarify the potential progenitor role of the fibroblast subtype, which is out of scope for this study. (2) IntFib (this study), Fib8 (McLellan) and interferon response fibroblast (IFN_r, Forte) were consistently associated with a strong gene expression related to an interferon response and marker by genes like *Ifit1* and *Ifit3*. We performed RNA in situ stainings for *Ifit1* and found local restricted areas of strong expression (new Extended Data Fig. 4b). The exact role of these interferon responsive fibroblast is currently unknown (Soliman et al. *Matrix Biol* 2020 PMID: 32446910). (3) ECM-Fib, FibThbs4 and matrifibrocytes (MFC) showed high correlation in our analysis. We think these fibroblast are of major importance for tissue remodeling in chronic injury, since they occurred only later after acute injury by MI. Furthermore we were able to show that ECM-Fib are highly injury related (new Fig. 3f) and express Periostin (new Fig. 3d), but not *Acta2* (new Extended Data Fig. 6f-g, i-j). Our correlation and integration analysis showed a clear difference between ECM-Fib and myofibroblasts, detected only after MI (new Extended Data Fig. 6b, d). Compared to fibroblast analyzed in previous approaches, we think periostin and FAP are rather broad markers of fibroblast activation, since they are upregulated in all fibroblast subtypes (new Fig. 4a).

To address these questions we have now also added the following sections to the results section of the revised manuscript:

*"To further analyze whether fibrotic ECM deposition is primarily driven by specific fibroblast subtypes, we compared ECM expression across fibroblast subtypes. The ECM score was largely similar in all fibroblast subpopulations in homeostasis, while after TAC specifically the ECM-Fib cluster showed the highest ECM score, followed by Fib1 and IntFib (Fig. 3g and Extended Data Fig. 4f). Comparing particular collagens including different functional groups of collagens (Karsdal et al. *Liver Int* 2020 PMID: 31997561), revealed Fib1 and ECM-Fib as highest collagen expressing fibroblasts (Fig. 3h and Extended Data Fig. 4g, h). ECM-Fib showed strong expression of fibrillar collagens (e.g. *Col1a1/2*) 14 days after TAC, which declined at 28 days. Fib1 on the other hand primarily expressed network forming (e.g. *Col4/6* subtypes) and multiplexin collagens (e.g. *Col15a1*) after injury."*

*"To verify this data with an orthogonal method and dissect if there are specific localizations of these identified fibroblast subtypes, we performed in situ hybridization of specific marker genes on left ventricular tissue. Separate co-stainings for *Ifit1*, *Fgl2* and *Atf3* with *tdTom*, confirmed the co-expression of these marker genes in *Gli1* or *Pdgfrβ* lineage derived cardiac fibroblasts (Extended Data Fig 4b). For *Ifit1*, we observed *tdTom* co-staining in focal interstitial areas (Extended Data Fig. 4b).*

Fgl2 and Atf3 were expressed in fibroblasts throughout the left ventricular tissue with no noticeable enrichment in areas of fibrosis or larger vessels (Extended Data Fig 4b). Atf3 has already been described as a marker gene for cellular stress response induced by different stimuli (Pakos-Zebrucka et al. EMBO Rep 2016 PMID: 27629041). More specifically within the heart, Atf3 was reported to be elevated towards the terminal stage of the mitochondrial integrated stress response, suggesting the Atf3-Fib subtype might be involved in this response (Forsström et al. Cell Metab 2019 PMID: 31523008)."

"We next compared our fibroblast subtypes to subtypes defined in the recently published datasets from Forte et al. (Cell Reports 2020 PMID: 32130914) and McLellan et al. (Circulation 2020 PMID 32795101) by correlation (Extended Data Fig. 6a, b). We were able to identify three conserved subtype clusters existing in all three datasets: (1) fibroblasts of high ECM expression: ECM-Fib, FibThbs4/FibCilp, matrifibrocytes (MFCs); (2) interferon responsive fibroblasts: IntFib, Fib8, interferon response fibroblast (IFNr); (3) Ly6a⁺/CD248⁺ fibroblasts: Fib2, Fib5, progenitor-like state fibroblast (PLS). Integration of the datasets validated this finding as correlating subtypes clustered together (Extended Data Fig. 6c, d). At this point it became apparent that one fibroblast subtype is missing from our dataset, due to dissimilarity in study design. Wif1⁺/Dkk3⁺ fibroblast seem to be located specifically in the region of heart valves, the part of the heart which in our study was only used for imaging, while Forte et al. and McLellan et al. used whole hearts (Extended Data Fig. 6e-h). Heart valve associated fibroblast, marked by Wif1/Dkk3 expression, have also been suggested as a separate subtype by Forte et al. (Cell Reports 2020 PMID: 32130914) and Muhl et al. (Muhl et al. Nat Commun 2020 PMID: 32769974). One other important observation of our integration approach was the different occurrence of Thbs4⁺ high ECM expressing fibroblasts and Acta2⁺ myofibroblasts, with the latter clearly found early after myocardial infarction (MI) (Forte et al. Cell Reports 2020 PMID: 32130914), though not in the TAC (our data) nor AngII model (McLellan et al. Circulation 2020 PMID 32795101)(Extended Data Fig. 6i-l). This suggests a strong diversity in injury dependent activation of fibroblasts. Myofibroblasts seem to be required mainly in acute ischemic injury with subsequent replacement fibrosis, since contractile myofibers might be needed to contract the large wounds after MI while they might not be required in interstitial fibrosis caused by increased afterload."

Rebuttal Fig. 3: *Atf3* expression in fibroblast datasets

UMAP representation of the fibroblast subtypes from this study, McLellan et al. and Forte et al. with cluster labels and gene expression for *Atf3*. Violin plots show the *Atf3* expression summarized per subcluster, public datasets combined with fibroblast from this study for better comparison.

4. Considering the role of proteases/antiproteases in cardiac remodeling and the potential for distinct roles of “matrix-degrading” vs “matrix-preserving” cells, it would be informative to ask whether some of the subpopulations identified by the authors exhibit distinct expression patterns of MMPs/TIMPs.

We agree and appreciate this suggestion. We have looked into the expression of MMPs and TIMPs in our fibroblast, EC and mural cell subtypes (Rebuttal Fig. 4). Interestingly, we found a significant upregulation of *Timp3* and *Timp4* in multiple EC subtypes. *Timp3* is described as a potent inhibitor of angiogenesis (Qi et al. Apoptosis 2015 PMID: 25558000) and therefore might be involved in the decreased proliferative activity of EC 28 days after TAC. For MMPs the overall coverage was rather low and we only observed an upregulation of *Mmp2* in fibroblasts and *Mmp14* pericytes (Rebuttal Fig. 4).

Rebuttal Fig. 4: Gene expression of all detected TIMP and MMP in fibroblast, endothelial and mural cells

Violin plot of the gene expression of all TIMP and MMP that were detected in the fibroblast, endothelial and mural cells, separate for each subcluster and condition (*: p<0.001, differential gene expression test by MAST compared TAC 14d or TAC 28d to Sham).

5.Regarding endothelial cell phenotype: the authors point out that “further analysis pointed towards failed angiogenesis”. How is this conclusion supported by the data? Did the authors observe capillary rarefaction?

We appreciate the concerns raised by the reviewer. We added a second time point 28 days after TAC for our *Cdh5CreER;tdTomato* lineage tracing experiment, to analyze changes in endothelial cells over time. A composition analysis of *Cdh5* fate traced cells comparing sham, TAC 14d and TAC 28d, showed first an increase in CyclEC and RepEC populations 14 days after TAC followed by a decrease at 28 days after TAC as compared to sham and TAC 14d (new Fig. 6g and new Extended Data Fig. 8e). This suggests an initial pro-angiogenic phase of the injured heart in response to increased oxygen and metabolic demand, which is not sustained longer as the number of CyclEC and RepEC is decreased 28 days after TAC. In addition to this we quantified proliferation of EC by KI-67 staining across *Cdh5CreER;tdTomato* mouse hearts from all time points (new Fig. 6h), which confirmed the finding of increased proliferative EC early after injury with a subsequent decrease at the later time point. RNA in situ hybridisation for the angiogenic EC marker gene *Apln* showed enriched expression of *tdtom*⁺ *Apln*⁺ endothelial cells in a region of *Col1a1*⁺ cardiac fibrosis in line with an angiogenic response of EC particularly in areas of injury (new Extended Data Fig. 8f). Even though we did not observe capillary rarefaction even 28 days after TAC (new Extended Data Fig. 8g), we were able to cover the transcriptomic response of EC after TAC induced injury. It remains unclear if the reduced proliferation of EC 28 days after TAC together with the detected transcriptional changes at this time point could already be an early sign of a capillary cardiomyocyte mismatch which has been postulated as contributor to cardiac hypertrophic remodeling and heart failure (Sano et al. Nature 2007 PMID:17334357; Mohammed and Redfield Circulation 2015 PMID:26481572). However, we would like to point out that our

differential gene expression analysis detected upregulation of genes related to blood vessel development and cell migration, with the changes beginning most prominent for the TAC 28 days time point (new Fig. 6k, l and new Extended Data Fig. 9e). This pro-angiogenic transcriptional profile in EC, which occurs 28 days after TAC, is in contrast to the observed reduced number of proliferating EC for this time point. Furthermore, TF activity prediction and pathway activity prediction revealed processes that might be responsible for a rather poor angiogenic potential in heart (new Fig. 6i, j and new Extended Data Fig. 9a, c), as it was recently reported by Kocijan et al. (Cardiovasc Res. 2021 PMID: 31999325). Revised receptor ligand analysis demonstrated a pattern of reduced signaling towards the EC niche (new Fig. 5h-i), which is potentially further involved in the reduced angiogenesis at the later time point. Considering our new analysis of EC we adjusted our conclusion with regards to failed angiogenesis.

We have now added the following paragraphs to our results:

"Composition analysis showed an increase in CyclEC and RepEC numbers at day 14 after TAC, with a subsequent decrease of both populations at the later time point at even significantly lower numbers as compared to sham (Fig. 6g and Extended Data Fig. 8e). This suggests an initial pro-angiogenic phase of the injured heart in response to increased oxygen and metabolic demand (Oka et al. Circ Res 2014 PMID: 24481846), which is not sustained at the later time point. Quantification of KI-67 tissue expression in Cdh5 lineage tracing hearts confirmed this finding (Fig. 6h). RNA in situ hybridization experiments for the angiogenic EC marker gene Apln showed enriched expression in a region of cardiac fibrosis, in line with an angiogenic response of EC particular in areas of injury (Extended Data Fig. 8f). Interestingly, while we observed a clear trend towards decreased capillary numbers this did not reach significance (Extended Data Fig. 8g). The observed decrease of proliferation EC after the initial phase might be related to the finding of a recent study, which demonstrated an abortive angiogenic response in the heart (Kocijan et al. Cardiovasc Res 2021 PMID: 31999325)."

"The significantly diminished proliferative EC populations at 28 days after TAC together with the detected transcriptional changes at this time point could contribute to the mismatch in capillary density to cardiomyocyte size, which is thought to be involved in cardiac hypertrophic remodeling and heart failure (Sano et al. Nature 2007 PMID: 17334357; Mohammed and Redfield et al. Circulation 2015 PMID: 25552356)."

6. A limitation of the study is the investigation of a single timepoint (2 weeks after TAC). Pressure overload results in a sequence of cellular and functional changes that cannot be recapitulated through studies of a single timepoint. The reviewer recognizes that the type of analysis performed here cannot be easily repeated at multiple timepoints. However, the authors need to discuss the limitations.

We agree that the investigation of a single time point will not capture all changes occurring in a pressure overload injury model. We have now added a second time point after 4 weeks for fibroblasts and endothelial cells, using *Gli1CreER;tdTomato* mice recombining in all fibroblasts we had identified and *Cdh5CreER;tdTomato* mice, to capture more changes over time. A systematic study of the TAC model by Platt and colleagues showed that within the first 4 weeks after the surgery most of the fibrotic remodeling occurs, together with strong hemodynamic changes (Platt et al Front Physiol. 2018 PMID: 29867532, Richards et al. Sci Rep. 2019 PMID: 30971724). However, after this earlier remodeling the function slowly declines to a decompensated state, which was not captured by our study. We added this limitation to our discussion. Regarding insights from data of the new

time point 28 days after TAC, we would like to highlight the following findings. For fibroblast, we found a significant continuous expansion of ECM-Fib, which was most prominent at the latest time point (new Fig. 3f and new Extended Data Fig. 4d). Furthermore we predicted increasing TEAD1 transcription factor activity and validated a pro-fibrotic role for TEAD1 *in vitro* (new Fig. 4f-j). RNA velocity analysis combined with trajectory analysis including the new data revealed a potential differentiation path from Fib1 to ECM-Fib (new Fig. 5 and new Extended Data Fig. 7c-g). For endothelial cells we found reduced proliferation towards 28 days after TAC (new Fig. 6g-h and new Extended Data Fig. 8e) and detected a larger amount of differentially expressed genes associated with angiogenesis, hypoxia signaling and cell migration (new Fig. 6k, l and new Extended Data Fig. 9e). Taken together, we improved our coverage of remodeling processes of the vascular niche towards later hypertrophic remodeling.

Nevertheless, we agree that the limited time points are one limitation of our study and we added the following part to our discussion:

"Since we were only able to generate a single early time point after TAC and no functional validation for mural cell contribution to cardiac scar formation, further studies are needed to validate their contribution to cardiac remodeling and fibrosis. Furthermore, we need to point out that the time frame of four weeks after TAC represents the early hypertrophic remodeling phase and not the later decompensated phase of HF (Platt et al. Front Physiol 2018 PMID: 29867532; Richards et al. Sci Rep 2019 PMID: 30971724)."

7. The authors state that the PDGFRbetaCre-derived population expanded following TAC; these cells exhibited fibroblast phenotype. Does this driver label both fibroblasts and mural cells? Does it label all fibroblasts?

We thank the reviewer for these two interesting questions.

Yes, we found that the *PdgfrβCreER*-driver labeled both mural cells and fibroblasts in the RNA sequencing data (Fig. 2a-d). Further, in the immunofluorescence stainings we also observed tdTomato positive cells in the hearts of *PdgfrβCreER* mice localized in the medial layer of arteries (VSCMCs) as well as in the adventitia of arteries and interstitially within the myocardium (Fig. 1f). The *PdgfrβCreER*-driver overlaps with the entire fibroblast population found in the dataset (new Fig. 3b-c and new Extended Data Fig. 4a). In our opinion the *PdgfrβCreER*-driver shows the overall broadest recombination in mesenchymal populations including fibroblasts, pericytes and vascular smooth muscle cells.

8. The abstract requires revisions and clarifications: a) "We identified that injury causes a loss of cellular cross-talk towards endothelium in the vascular niche". The meaning of this sentence is unclear. Please revise and be more specific. b) The final sentence of the abstract the authors state that a "previously unappreciated role of pericytes in cardiac remodeling and fibrosis was revealed". However, the rest of the abstract does not include description of data supporting this conclusion. Please add a couple of sentences that support this final conclusion.

We appreciate the reviewer's concern and changed our abstract accordingly. The first statement was previously based on our observation of a reduced signaling of fibroblast towards endothelial cells in our Ligand-Receptor interaction analysis. We revisited this analysis after adding a second time point 28 days after TAC. Our new

results are more focused on the major subtypes of fibroblast, endothelial cells and mural cells. Here, we found the overall signaling towards capillary, capillary vein, capillary artery and angiogenic EC to be reduced after TAC (new Fig. 5g-i). Nevertheless, we removed this statement from the abstract, since we decided to report more important findings from our new data.

The second statement about the role of pericytes was also removed from the abstract, since we came to the conclusion that our data does not sufficiently support the statement. We have also toned down the contribution of pericytes to cardiac scar formation in the text as discussed also above (answer to question 1).

Reviewer #3 (Remarks to the Author):

The manuscript by Peisker and Kramann and colleagues probes non-cardiomyocyte cell heterogeneity in a murine model of heart failure due to pressure overload (transaortic constriction; TAC) using single cell transcriptomics. This work was facilitated by cell sub-population enrichment using Cre-based cell marking in sham and TAC conditions using a variety of Cre strains generally specific for fibroblasts, endothelial cells and mural cells. Cre strains were also used to highlight and clarify the spatial distribution of marked cells and the apparently rare deviations from baseline cell fates using immunostaining. The choice of the TAC model and 2-week timepoint allows for the development of pressure-overload heart failure that may resemble the impact of hypertension in humans, adding to the myocardial infarction and AngII-induced hypertension single cell datasets already published, albeit that hypertension in humans would be a chronic disease developing over years if not decades. Thus, whereas TAC is a commonly-used model, its relationship to human high blood pressure is debatable. Furthermore, results may be expected to be similar to the AngII model.

In general, this is a descriptive paper that would nonetheless add significant new data to previous sc/snRNAseq analyses of cardiac homeostasis and disease.

We thank the reviewer for highlighting the contribution of our data to the understanding of cardiac diseases and have addressed the reviewer's concerns to the best of our abilities, as well as added certain limitations of our study to our conclusions.

Major Issues:

1. A strength of the paper is the enrichment for interstitial cells subtypes before scRNAseq; however, this comes at the cost of having only a single time point, thus the power to resolve earlier cell states and cell trajectories is markedly diminished.

We appreciate the concern of the reviewer regarding the analyzed single time point and added a second time point for fibroblast and endothelial cells to extend our analysis to changes that occur later in the remodeling process. Unfortunately, we were not able to generate a later time point for the mural cell data and added this limitation to the discussion. Our new datasets for fibroblast and endothelial cells 28 days after TAC allowed us to analyze transcriptome changes occurring later in hypertrophic remodeling.

For fibroblast we found ECM-Fib to be expanding even further (new Fig. 3f) and overall the transcriptomic response of fibroblast was found more prominent (new Fig. 4a). Similar to finding from our earlier time point we detected TEAD1 activity as highly upregulated after 28 days TAC (new Fig. 4f-g). The trajectory analysis including the later time point further pointed towards a differentiation from Fib1 to ECM-Fib (new Fig. 5).

For endothelial cells we observed a decrease of proliferation towards the later time point (new Fig. 6g, h) and a more prominent transcriptomic response overall (new Fig. 6k). Interestingly this response was highly associated with processes of cell migration for arterial endothelial cells (new Fig. 6k, l and new Extended Data Fig. 9e).

We have further added the limited time points as a limitation of our study and added the following sentences:

“Our trajectory interference analysis suggested a more general pool of fibroblasts as potential progenitors, mostly consisting of the Fib1 type, however, this computational model needs to be considered carefully. The progenitor role of cardiac fibroblasts remains controversial and more experiments are needed to dissect the progenitor potential of the different cardiac subtypes.”

“Since we were only able to generate a single early time point after TAC and no functional validation for mural cell contribution to cardiac scar formation, further studies are needed to validate their contribution to cardiac remodeling and fibrosis. Furthermore, we need to point out that the time frame of four weeks after TAC represents the early hypertrophic remodeling phase and not the later decompensated phase of HF (Platt et al. Front Physiol 2018 PMID: 29867532; Richards et al. Sci Rep 2019 PMID: 30971724).”

2. The authors embark on a detailed lineage marking exercise that allows cell purification (with retrospective identification of marked cells by scRNAseq) and tracing of lineage descendants; however, there is no clear aim or model provided for this section. This is a missed opportunity. Given that much has been made of the potential plasticity of stromal cells, clear goals and outcomes (and limitations) would strengthen this paper.

We appreciate the concern of the reviewer and adjusted our manuscript accordingly. We defined clear goals in the introduction, described outcomes and added limitations through the manuscripts. The following limitation are included: We focused our single cell analysis on the earlier phase of hypertrophic remodeling within the first four weeks after TAC surgery, thereby missing out on later phases of progressive decompensation. For mural cells we only generated data for the first two weeks and overall observed a milder phenotype in mice contributing to these cells.

Due to our experimental design with the goal to generate single cell sequencing and imaging data from the same heart, we missed out on a population of *Wif1*⁺ fibroblast, which are associated with the heart valves. Furthermore we added a combined analysis of our single cell data from fibroblast with the data generated by Forte et al. (Cell Reports 2020 PMID: 32130914) and McLellan et al. (Circulation 2020 PMID 32795101), which is based on digestion of whole hearts (new Extended Data Fig. 6). From the results of the comparison we were able to define a population of fibroblast with *Wif1* and *Dkk3* as highly expressed markers (new Extended Data Fig. 6e-g), which we did not cover at substantial amounts in our study. There is strong evidence in the literature that these *Wif1*-fibroblasts

represent a population of ventricular fibroblasts that is local enrichment towards the heart valves (new Extended Data Fig. 6h) (Forte et al. Cell Reports 2020 PMID: 32130914; Muhl et al. Nat Commun 2020 PMID: 32769974; Inoue et al. J Mol Histol 2017 PMID: 27866302). We are aware of this limitation regarding the completeness of our coverage of cardiac fibroblast heterogeneity.

Similar to this, our lineage tracing approach with stringent sorting for tdTom⁺ cells excluded cardiomyocytes and immune cells, as two major cardiac cell types. Therefore we are not able to include these cell types in our ligand receptor interaction analysis with the vascular niche.

We added the following text to our manuscript to outline our goals, outcomes and limitations better:

“Various studies have demonstrated plasticity of stromal and endothelial cells particularly in disease (Chong et al. cell stem cell 2011 PMID: 22136928, Lovisa et al. Nature med. 2015 PMID: 26236991, Nosedá et al. Circ J. 2015 PMID: 26073608, Tombor et al. Nat. Commun 2021 PMID: 33514719; Zeisberg et al. Nat Med 2007 PMID: 17660828). However, due to the strong heterogeneity of the vascular niche, the individual contributions of different vascular and perivascular cell types to cardiac remodeling remain unclear. The goal of this study was to dissect the heterogeneity and individual contribution of stromal and endothelial cell types to cardiac remodeling in heart failure by combining inducible genetic fate tracing with single cell RNA-sequencing and confocal imaging. For this we used various mouse lines for genetic fate tracing of cardiac endothelial, pericyte, VSMC and fibroblast lineages. This approach enabled us to capture and map cell type specific transcriptomic changes over two independent time points in hypertrophic left ventricular remodeling. Our data indicates that all cardiac fibroblast populations shift towards an extracellular matrix expressing profile after injury. However, one specific fibroblast subtype marked by Thbs4 expression showed significant expansion, likely differentiated from a broader pool of fibroblasts and expressed distinct ECM composition. Endothelial cells responded early after injury by increased proliferation, but failed to sustain angiogenesis in later remodeling. Taken together, we provide an extensive overview of cell type specific transcriptomic changes in the cardiac vascular niche in hypertrophic remodeling.”

“Our trajectory interference analysis suggested a more general pool of fibroblasts as potential progenitors, mostly consisting of the Fib1 type, however, this computational model needs to be considered carefully. The progenitor role of cardiac fibroblasts remains controversial and more experiments are needed to dissect the progenitor potential of the different cardiac subtypes.”

“Since we were only able to generate a single early time point after TAC and no functional validation for mural cell contribution to cardiac scar formation, further studies are needed to validate their contribution to cardiac remodeling and fibrosis. Furthermore, we need to point out that the time frame of four weeks after TAC represents the early hypertrophic remodeling phase and not the later decompensated phase of HF (Platt et al. Front Physiol 2018 PMID: 29867532; Richards et al. Sci Rep 2019 PMID: 30971724). One potential limitation of our study is that in the PdgfrβCre and Myh11Cre crossings we did not see significant hypertrophy suggesting a milder phenotype and or variance in the model. Additionally, we acknowledge that our lineage tracing based enrichment strategy excluded cardiomyocytes, therefore we lack analysis of the potential influence from this cell population.”

3. In Figure 1C, it is apparent that not all Cre crosses showed significant hypertrophic changes – specifically Pdgfrb-Cre and Myh11-Cre. In others, significance seems to be driven by a few examples. Given the 2-week time point when TAC hearts should be showing hypertrophy (increased heart weight to tibial length), this suggests technical variation in the TAC surgery, which could add considerable noise to the data and raising concerns about analysis of population

proportions. The authors should have checked the pressures on either side of the constriction.

We agree with the concerns raised by the reviewer regarding technical variation of our TAC model. We would like to point out that even though the hypertrophic changes were not significant for *Pdgfrb β Cre* and *Myh11Cre*, we still observed a trend toward hypertrophy and we found prominent histological changes by our tissue imaging. It is true that some mice seem to develop a more severe disease than others. Unfortunately, we did not measure pressures on either side of the constriction in the initial set of mice, however, in the new mouse groups we have added this (new Extended Data Fig. 1a) showing increased peak trans-TAC pressure after surgery. We furthermore provide echocardiographic data for these new mice showing reduced ejection fraction.

We added the following sentence as a limitation of our study:

*"One potential limitation of our study is that in the *Pdgfrb β Cre* and *Myh11Cre* crossings we did not see significant hypertrophy suggesting a milder phenotype and or variance in the model."*

4. For the statistical testing of differences in population proportions between sham and TAC (e.g. Fig 1g), have the authors tried a differential proportion test developed for single-cell data, e.g. scDC (Cao et al. BMC Bioinformatics 2019) or DPA (Farbehi et al., eLife 2019). A standard contingency table test will not be able to account for the inherent technical variability in scRNA-seq data.

We agree with the concerns raised by the reviewer and changed our composition analysis. We used the `scProportionTest` package (<https://github.com/rpolICASTRO/scProportionTest>), which was designed to test proportional changes between cluster composition by permutation test via bootstrapping. We provided detailed plots for the results of all comparisons in our extended data figures for all subclusters (new Extended Data Fig. 4d, 8e, 10f).

5. There are a high number of samples processed, requiring batch correction. It appears that the challenges of data integration were only considered retrospectively: "...it became apparent to us that the integration of all 12 data sets requires a different strategy than integrating two or six datasets."

We regret that this sentence in the paper suggests that we only considered batch correction after the experimental procedure. We would like to state that bioinformatic analysis is a rapidly evolving field; new integration strategies are sprouting at a high frequency and distinct methods offer a trade-off between removal of batch effects and maintenance of biological signal (Luecken and Theis Mol Syst Biol 2019 PMID: 31217225). The integration of 14 individual single cell libraries based on different cre drivers provided a special challenge for integration, since there is a distinct degree of cell type overlap between cre drives and some of the cre drivers have no overlap of cells (e.g. *Cdh5CreER* and *Gli1CreER*). We therefore applied the method called "harmony" for the integration of all samples. Harmony outperformed other integration methods in multiple recent benchmarking studies (Tran et al. Genome Biol. 2020 PMID: 31948481; Chazarra-Gil et al. Nucleic Acids Res. 2021 PMID: 33524142). Of note, these studies also

highlight the fact that each integration approach requires thorough consideration of the method used, followed by careful examination of the results, i.e. major groups have markers associated with a particular cell type. Moreover, we were able to validate several of the findings from our data integration by stainings or *in vitro* experiments. Therefore we conclude that our data integration strategy was successful to uncover the heterogeneity of stromal cells in the heart.

6. One of the puzzling outcomes of this paper is the high similarity between sham and TAC samples. Only one population, MatriF, expresses Periostin, a marker of activated fibroblasts and myofibroblasts. This is also the only population to express Thbs4. It seems incongruous that these are present in abundance in Sham samples and increase to a fairly modest degree after TAC and I suspect that this may be due to technical challenges of batch correction. The authors use Seurat and Harmony batch correction methods to combine their datasets. Given that they are correcting between sham and TAC, there may be a risk for 'over-correction' such that cell states that should be specific to TAC might end up observed in sham or vice versa, thus confusing the biological interpretation of the data. Indeed, as noted, among fibroblasts about 40% of the MatriF cells are in sham condition, which seems improbable given findings from other scRNA-seq studies comparing fibroblasts in sham and diseased hearts (e.g. MacLellan et al., Circulation 2020, Forte et al. Cell Reports 2020, Farbehi et al., eLife 2019). In MacLellan et al., which may be the most relevant to this study, their Thbs4-fibroblast population (which may be similar to the author's MatriF cells) is essentially unique to their diseased condition. The authors should consider the impact of batch correction on interpretation and compare to results obtained from not correcting between sham and TAC (e.g. analysing sham and TAC separately).

We appreciate the concern by the reviewer. To give a clear answer to this question, it is very important to define what a subset is. Transcriptional changes occur in all fibroblasts after TAC, but it remains difficult to draw the line between a change in the subtype state or true differentiation towards a different subtype. Our data indeed suggests that the ECM-Fib (renamed from previous "MatriF") are already present in non-injured hearts. As suggested by the reviewer, we analyzed our sham datasets separate from the TAC datasets (Rebuttal Fig. 2). Unsupervised clustering of only sham fibroblasts revealed a small, but distinct cluster of *Postn* high expression cells (Rebuttal Fig. 2a, upper UMAP and violin plot). Transferring of the cluster annotation from the full integration of all fibroblasts to the sham subset, showed that the *Postn* high fibroblasts are part of the ECM-Fib cluster in the full integration (Rebuttal Fig. 2a, lower UMAP and violin plot). We performed the same analysis for the control datasets from McLellan et al. (Circulation 2020 PMID 32795101) and Forte et al. (Cell Reports 2020 PMID: 32130914) (Rebuttal Fig. 2b, c), with the same result. Both public control datasets contain a small number of *Postn* high fibroblasts, which are later clustering together with injury related subtypes (FibCilp and FibThbs4 for McLellan; Myof and MFC for Forte). Of note, the cell number of fibroblast in the two public control datasets was considerably lower, then in our sham dataset. This is most likely due to the enrichment strategy of our study, which covered more ECM-Fib from our sham hearts by sorting of tdTomato⁺ cells after recombination using mesenchymal Cre drivers. We also added a new figure part to the main manuscript to clarify this topic (new Extended Data Fig. 7a). Furthermore, we also performed RNA in situ hybridisation for *Postn* in non-injured (sham) murine hearts and demonstrated the

presence of *tdtom⁺ Postn* expressing fibroblasts in homeostasis (new Extended Data Fig. 7b). We think this is sufficient evidence for the presence of a small non-activated ECM-Fib population cardiac in homeostasis.

After we added two new samples to the datasets (28days TAC *Gli1CreER* and *Cdh5CreER*) we revisited our integration and filtering strategy (Extended Data Fig. 2a). As shown in the rebuttal figure 5a below, integration of our fibroblast data without any batch correction, led to batch driven clustering, where the different lineage tracing genotypes tend to form separate clusters. Therefore, we decided to use the integration method from *seurat*, since this method is widely accepted for scRNA data integration. We also tested harmony based batch correction separately on our fibroblasts samples and found again a subcluster with *Thbs4* expression after TAC, which also included cells from the Sham conditions (Rebuttal Fig. 5b-c). We hypothesize that our enrichment strategy was more efficient to enrich these cells from sham hearts, compared to the whole heart approaches of McLellan et al. and Forte et al.

In addition, we would like to point out that our correlation analysis with the subclusters of McLellan et al. found our ECM-Fib to be highly similar to *Thbs4*-Fib and *Cilp*-Fib in their study, so both clusters must probably be counted together in comparison. Besides this, there was no specific batch correction method used in the McLellan study, which could lead to *Thbs4*-Fib and *Cilp*-Fib clustering more separate from the control cells due to batch effects. In conclusion, we agree with the reviewer that the impact of batch correction needs to be considered carefully and we considered this while interpreting our results.

Rebuttal Fig. 5: Fibroblasts batch effect and harmony batch correction

a, UMAP embedding of all fibroblast separate for each lineage tracing genotype and condition. No specific batch correction method was applied. **b**, UMAP embedding of all fibroblast separate for each lineage tracing genotype and condition, after applying a batch correction with the function from the R package harmony. **c**, UMAP of the harmony corrected fibroblasts with clustering. **d**, Barplot of the composition per cluster, showing the normalized contribution of cells from the different conditions to the clusters. **e**, Violin plot of the *Thbs4* gene expression in each cluster, separate for each condition.

7. The use of the term MatriF to describe the Periostin+/Thbs4+ fibroblast subset is confusing given its similarity to matrifibrocytes, which I do not think features in this model, but are a prominent sub-state in the MI model. Nomenclature is going to be tricky in the single cell field without unintentional distractions; thus I suggest choosing an alternate name.

We thank the reviewer for aiding us in selecting the appropriate nomenclature for our clusters. We agree with the reviewer that a robust nomenclature for single cell data should be established and conformed to. We agree that definition of subtypes and substages, remains a major challenge for the single cell field. We adjusted MatriF to ECM-Fib, as these fibroblast are chief ECM producers and the name is more distinguishable from "matrifibrocytes". However, our comparison to the fibroblast datasets of Forte et al. and McLellan et al. revealed three potentially

conserved fibroblast subtypes (new Extended Data Fig. 6), including ECM-Fib, which are highly similar to Thbs4-Fib or the McLellan study and matrifibrocytes (MFC) of the Forte study. It seems plausible that these subtypes in fact, are all the same subtype, which occurs in non-acute cardiac remodeling and is characterized by higher expression of ECM related genes. We think future studies will need to clarify if ECM-Fib, Thbs4-Fib and MFC represent different fibroblast subtypes, which occur in an injury specific manner, or if they are the same.

We added the following parts to the discussion:

“Our data confirms that fibroblasts are the major source of fibrosis in the murine heart following pressure overload⁴. Furthermore, all fibroblast subtypes seem to get activated and increase their ECM gene expression contributing to scar formation, however we identified ECM-Fib as the highest ECM gene expressing population. We think subtypes similar to ECM-Fib were found by previous single cell sequencing studies. McLellan et al. describe FibThbs4 and FibCilp as injury related subtypes with increased ECM remodeling related gene expression. Forte et al. found a fibroblast subtype that occurred only at later stages of cardiac remodeling 14 days after MI, which they termed matrifibrocytes according to an earlier described fibroblast type (Fu et al. J Clin Invest 2018 PMID: 29664017), found in the maturing scar. Taken together, it seems plausible that these results all describe a similar subtype of fibroblasts, which occurs in non-acute cardiac remodeling and is characterized by higher expression of ECM related genes.

We did not observe specific expression of Acta2 for the ECM-Fib subtype or any other fibroblast subtype in our study even when integrating with data from other murine heart disease models^{15,23}, indicating that in our injury model no classic myofibroblasts developed. We propose that depending on the type of cardiac injury the requirement of contractile proteins is needed in scar forming myofibroblasts. This might be particularly important following MI to quickly close the large defect of the left ventricular wall in replacement fibrosis while in interstitial fibrosis following pressure overload this might not be an important feature of myofibroblasts. For many years the term myofibroblast was used for the major matrix producing, fibrosis driving cell types. However, data presented here and various other scRNA-seq datasets across major organs (Kuppe et al. Nature 2020 PMID: 33176333; Henderson et al. 2020 Nature PMID: 33239795; Fu et al. J Clin Invest 2018 PMID: 29664017; Forte et al. Cell Reports 2020 PMID: 32130914; McLellan et al. Circulation 2020 PMID 32795101) indicated that the major matrix producing cell types are often not expressing classical myofibers (Acta2) and thus the terminology needs to be adapted.”

8. The authors report numerous stressed cell types in their lineage-traced datasets. Consistently, these populations appear to be represented more in sham than TAC, indicating that the stressed states are likely to be due to technical rather than biological factors. In this case, the stressed cells should be removed from the data prior to other analysis such as differential expression.

Regarding clustered termed “stressed” we tried to dissect if their transcriptomic profile is solely artificial due to the cell isolation. We performed staining with RNA *in situ* hybridization, comparison to publicly available dataset and a focused literature research. We found *Atf3* as the highest expressed marker gene in the corresponding fibroblast subcluster (new Fig. 3d) and we were able to detect *Atf3* expressing fibroblasts in cardiac tissue by RNA *in situ* hybridization (new Extended Data Fig. 4b), as well as *Atf3*⁺ endothelial cells (Rebuttal Fig. 6) and *Atf3*⁺ mural cells (new Extended Data Fig. 10c). The comparison of our data with the public datasets from Forte et al. and McLellan et al. revealed *Atf3* expressing fibroblast subclusters (Rebuttal Fig. 3), even though a specific cluster of *Atf3* expressing fibroblasts was not described there. Furthermore, a recent publication by

Forsström et al. (Cell metabolism 2019 PMID: 31523008) described *Atf3* in myoblasts to be activated towards the terminal stage of mitochondrial integrated stress response. The integrated stress response is a complex cellular program, which can get activated by different conditions and result in cell survival or cell death, however *Atf3* is described to be involved in the cellular survival response (Pakos-Zebrucka et al. EMBO Rep. 2016 PMID: 27629041). Here we would like to point out that filtering for apoptotic or in general dying cells in single cell RNA sequencing data is a standard procedure. Usually the percentual mitochondrial RNA (%-mt) content in those cells is higher than in the general population (Luecken and Theis Mol Syst Biol 2019 PMID: 31217225). We used a stringent cut off value of >6% to include only cells with a low %-mt content and exclude dying and low quality cells. Taken together, we provide evidence that the *Atf3* subtypes represent true transcriptomic heterogeneity of the respective cell types and not solely artificial profiles. Therefore we decided to not exclude these subclusters from our data. However, we can not exclude that these subtypes were more susceptible to transcriptomic changes by the isolation methods used by us and others. Therefore we did not perform more specific analysis on these subtypes.

Rebuttal Fig. 6: *Atf3*⁺ cardiac endothelial cells

Representative pictures of RNA ISH staining for *Atf3*, *Col1a1* and *tdTomato* on *Cdh5CreER;tdtomato* heart tissue.

9. Pertinent to points 7 and 8, the authors neglect to compare their data with previous cardiac scRNA-seq studies, which includes one study on an AngII hypertension model. Again, this is a missed opportunity and further complicates the field with an additional boutique nomenclature. As noted, there is the potential for technical artifacts in batch-correction that skew cell states - obviously the authors should address this possibility first.

We appreciate the reviewers suggestion to improve our study by comparing it more extensively to the current literature. Therefore, we performed comparison of our data sets with the public data sets from Forte et al. and McLellan et al., which provides further evidence for the fibroblast heterogeneity of the murine heart (new Extended Data Fig. 6). First, we compared our fibroblast subtypes with subtypes defined in the respective studies by correlation analysis (new Extended Data Fig. 6a). This lead to the identification of at least three conserved fibroblast subtypes, which seem be covered by all three datasets. (1) Fib2 in our study were highly similar to Fib5 of the McLellan study and progenitor-like state (PLS) fibroblast characterized in the study be Forte. Conserved marker genes of this cluster are *Cd248*, *Ly6a* and *Pi16*. There are multiple recent studies suggesting fibroblast with

Ly6a expression as progenitor population (Tang et al. Circulation 2018 PMID: 30566021; Soliman et al. Cell Stem Cell 2020 PMID: 31978365; Buechler et al. Nature 2021 PMID: 33981032). We were not able to find any transcriptomic feature for this population indicating a specific progenitor function, however a more focused experimental design will be necessary to clarify the potential progenitor role of the fibroblast subtype, which is out of scope for this study. (2) IntFib (this study), Fib8 (McLellan) and interferon response fibroblast (IFNr, Forte) were consistently associated with a strong gene expression related to an interferon response and marker by genes like *Ifit1* and *Ifit3*. We performed RNA in situ stainings for *Ifit1* and found local restricted areas of strong expression (new Extended Data Fig. 4b). The exact role of these interferon responsive fibroblast is currently known (Soliman et al. Matrix Biol 2020 PMID: 32446910). (3) ECM-Fib, FibThbs4 and matrifibrocytes (MFC) showed high correlation in our analysis. We think these fibroblast are of major importance for tissue remodeling in chronic injury, since they occurred only later after acute injury by MI. Furthermore we were able to show that ECM-Fib are highly injury related (new Fig. 3f) and express Periostin (new Fig. 3d), but not *Acta2* (new Extended Data Fig. 6f-g, i-j). Our correlation and integration analysis showed a clear difference between ECM-Fib and myofibroblasts, which were detected only after MI (new Extended Data Fig. 6b, d, i, j).

We added the following sections for the topic to our results and discussion:

"We next compared our fibroblast subtypes to subtypes defined in the recently published datasets from Forte et al. (Cell Reports 2020 PMID: 32130914) and McLellan et al. (Circulation 2020 PMID 32795101) by correlation (Extended Data Fig. 6a, b). We were able to identify three conserved subtype clusters existing in all three datasets: (1) fibroblasts of high ECM expression: ECM-Fib, FibThbs4/FibCilp, matrifibrocytes (MFCs); (2) interferon responsive fibroblasts: IntFib, Fib8, interferon response fibroblast (IFNr); (3) Ly6a⁺/CD248⁺ fibroblasts: Fib2, Fib5, progenitor-like state fibroblast (PLS). Integration of the datasets validated this finding as correlating subtypes clustered together (Extended Data Fig. 6c, d). At this point it became apparent that one fibroblast subtype is missing from our dataset, due to dissimilarity in study design. Wif1⁺/Dkk3⁺ fibroblast seem to be located specifically in the region of heart valves, the part of the heart which in our study was only used for imaging, while Forte et al. and McLellan et al. used whole hearts (Extended Data Fig. 6e-h). Heart valve associated fibroblast, marked by Wif1/Dkk3 expression, have also been suggested as a separate subtype by Forte et al. (Cell Reports 2020 PMID: 32130914) and Muhl et al. (Muhl et al. Nat Commun 2020 PMID: 32769974). One other important observation of our integration approach was the different occurrence of Thbs4⁺ high ECM expressing fibroblasts and Acta2⁺ myofibroblasts, with the latter clearly found early after myocardial infarction (MI) (Forte et al. Cell Reports 2020 PMID: 32130914), though not in the TAC (our data) nor AngII model (McLellan et al. Circulation 2020 PMID 32795101)(Extended Data Fig. 6i-l). This suggests a strong diversity in injury dependent activation of fibroblasts. Myofibroblasts seem to be required mainly in acute ischemic injury with subsequent replacement fibrosis, since contractile myofibers might be needed to contract the large wounds after MI while they might not be required in interstitial fibrosis caused by increased afterload."

"Our data confirms that fibroblasts are the major source of fibrosis in the murine heart following pressure overload⁴. Furthermore, all fibroblast subtypes seem to get activated and increase their ECM gene expression contributing to scar formation, however we identified ECM-Fib as the highest ECM gene expressing population. We think subtypes similar to ECM-Fib were found by previous single cell sequencing studies. McLellan et al. describe FibThbs4 and FibCilp as injury related subtypes with increased ECM remodeling related gene expression. Forte et al. found a fibroblast subtype that occurred only at later stages of cardiac remodeling 14 days after MI, which they termed matrifibrocytes according to an earlier described fibroblast type (Fu et al. J Clin Invest 2018 PMID: 29664017), found in the maturing scar. Taken together, it seems plausible that these results all describe a similar subtype

of fibroblasts, which occurs in non-acute cardiac remodeling and is characterized by higher expression of ECM related genes.

We did not observe specific expression of *Acta2* for the ECM-Fib subtype or any other fibroblast subtype in our study even when integrating with data from other murine heart disease models^{15,23}, indicating that in our injury model no classic myofibroblasts developed. We propose that depending on the type of cardiac injury the requirement of contractile proteins is needed in scar forming myofibroblasts. This might be particularly important following MI to quickly close the large defect of the left ventricular wall in replacement fibrosis while in interstitial fibrosis following pressure overload this might not be an important feature of myofibroblasts. For many years the term myofibroblast was used for the major matrix producing, fibrosis driving cell types. However, data presented here and various other scRNA-seq datasets across major organs (Kuppe et al. Nature 2020 PMID: 33176333; Henderson et al. 2020 Nature PMID: 33239795; Fu et al. J Clin Invest 2018 PMID: 29664017; Forte et al. Cell Reports 2020 PMID: 32130914; McLellan et al. Circulation 2020 PMID 32795101) indicated that the major matrix producing cell types are often not expressing classical myofibers (*Acta2*) and thus the terminology needs to be adapted.”

10. The authors raise a question in Discussion about the potential for MSC-like cells among fibroblasts. This is an issue that has been discussed previously in scRNA-seq studies focussed on cardiac fibroblasts, with MSC or progenitor-like fibroblast sub-populations reported (Forte et al. Cell Reports 2020, Farbehi et al., eLife 2019; Soliman et al., Cell Stem Cell 2020). The authors should include these previous findings in their discussion of MSC-like fibroblasts and in general better contextualise their work in light of previous papers in this field.

We thank the reviewer for this suggestion and agree. With regards to the MSC-like features of fibroblasts, we showed in the past that cardiac *Gli1*⁺ cells possess mesenchymal stem cell (MSC) characteristics in the dish (Kramann et al. Cell Stem Cell 2015 PMID: 25465115). Since our sequencing data from *Gli1CerER;tdtomato* lineage traced cells suggests, this population includes the full heterogeneity of cardiac fibroblasts, it remains unclear, if all *Gli1*⁺ cells have MSC-like progenitor features or just a subfraction.

Soliman et al. (Cell Stem Cell 2020 PMID: 31978365) reports a distinct heterogeneity of cardiac fibroblast, of which a large population of PDGFR α ⁺/SCA-1⁺ cells display progenitor features with “*in vitro* clonogenicity and *in vivo* and *in vitro* lineage potential”. These progenitors generate PDGFR α ⁺/SCA-1⁻ fibroblast upon injury. Our single cell data identified two different fibroblast populations with enriched *Sca-1* (gene symbol: *Ly6a*) expression Fib2 and IntFib (new Fig. 3d). Both subtypes are conserved across different studies as they were also found by Forte et al. and McLellan et al. (new Extended Data Fig. 6). In contrast to the loss of PDGFR α ⁺/SCA-1⁺ fibroblasts after injury as reported by Soliman et al., *Sca1* high Fib2 did not decrease after injury in our study. Similar findings were reported by Forte et al. and McLellan et al. The corresponding *Sca1* high population of progenitor-like state fibroblast (PLS) described by Forte et al. “[...] were relatively stable across all time points”. Interestingly, a lineage tracing approach with a *Sca1-2A-CreER* mouse by Tang et al., reported *Sca1*-traced PDGFR α ⁺ cell expansion in the injury region after MI, with an increasing number from remote to infarct zone (Tang et al. Circulation 2018 PMID: 30566021). In our opinion, the role of SCA-1⁺ fibroblasts remains controversial and more extensive lineage tracing studies, potentially based on the new *Sca1-2A-CreER* line, are needed to dissect the full potential of the fibroblast subtype.

We added the following more detailed section to the discussion.

“We have previously reported that Gli1 marks cardiac myofibroblast precursors that possess mesenchymal stem cell (MSC) characteristics in the dish and that genetic ablation of these cells ameliorates cardiac fibrosis and stabilizes cardiac function after ascending aortic constriction (Kramann et al. Cell Stem Cell 2015 PMID: 25465115). Our scRNA-seq data here indicates that Gli1CreER mice recombine in the various cardiac fibroblast populations. This raises the question whether all cardiac fibroblasts would possess potential MSC characteristics in the dish and whether a difference between fibroblasts and MSC-like cells exists.

A recent study by Soliman et al. reports a distinct heterogeneity of cardiac fibroblast, where a large population of PDGFR α ⁺/SCA-1⁺ cells display MSC-like progenitor features (Soliman et al. Cell Stem Cell 2020 PMID: 31978365). These progenitors generate PDGFR α ⁺/SCA-1⁻ fibroblast upon injury. Our single cell data identified two different fibroblast populations with enriched Sca-1 (gene symbol: Ly6a) expression, Fib2 and IntFib. Both subtypes seem to be conserved independent of injury models as they were also found by Forte et al. and McLellan et al.. In contrast to the loss of PDGFR α ⁺/SCA-1⁺ fibroblasts after injury as reported by Soliman et al., Sca-1 high Fib2 did not decrease after injury in our study and others (Forte et al. Cell Reports 2020 PMID: 32130914; McLellan et al. Circulation 2020 PMID 32795101). Interestingly, Tang et al. reported that Sca-1 fate traced PDGFR α ⁺ cells expand in the injury region after MI (Tang et al. Circulation 2018 PMID: 30566021). Our trajectory interference analysis suggested a more general pool of fibroblasts as potential progenitors, mostly consisting of the Fib1 type, however, this computational model needs to be considered carefully. The progenitor role of cardiac fibroblasts remains controversial and more experiments are needed to dissect the progenitor potential of the different cardiac subtypes.”

Other comments:

1. There are numerous spelling and grammatical errors throughout the manuscript, please check carefully.
We apologize and have adjusted grammar and spelling throughout the manuscript.
2. Reference to the model in McLellan et al. 2020 as myocardial infarction should be corrected.
We apologize for the mistake and have corrected it.

REVIEWERS' COMMENTS

Reviewer #1 (Remarks to the Author):

The authors have responded my qu\rtions and the majority of complex issues raised by other reviewers. I believe that the manuscript remains descriptive but it certainly adds important new expressional information to generate new hypotheses.

Reviewer #2 (Remarks to the Author):

The authors have addressed my main concerns. I have no further recommendations.

Reviewer #3 (Remarks to the Author):

Please see my comments below. I am happy to support publication but am rather on the fence about it. Other weaknesses not described below are the obviousness and lack of novelty of many findings, the general lack of new biological insights or hypotheses and general failure to depart from the ingrained model of fibrosis as a general ramping up (and down) of ECM gene expression. Same for the sections on ECs and VSMCs.

Overall, there are some useful comments and findings, strengthened by the use of multiple Cre lines and ability in the end to collapse different lineage subtypes down to a large pool of cells for comparative study. I see it more as an atlas than an insightful paper.

**

The authors have added substantially to the previous draft and in particular have added a later time point for the TAC model. Much of the draft has been rewritten!

Weaknesses still remain in this paper. 1. I don't see why the authors could not have used their VSMC/pericyte Cre drivers to ask to what extent mural cells contribute to ECM-Fib in this TAC model. 2. I disagree that the time points chosen represent early phases of pressure overload. EF is already substantially reduced at 2 weeks. There are several implication, among them whether addition of the late time point adds very much. The main finding is the ECM-Fib is increased from day0-14 and day14-28, making it the main focus of attention among fibroblasts. However, I wonder whether the trajectory analysis is meaningful. Fig. 5a is really not compelling if the claim is the trajectories indicate formation of ECM-Fib from Fib1. The same goes for the associated supplementary figure. I do not know how the authors decided to place heavy emphasis on this as opposed to alternative hypotheses (if earlier time points might be included). I cannot comment on how the monocle was set up as in Fig. 5c,d so as to focus on this transition; however, I suggest that this is focus is not justified based on results in Fig.5a. In any case, the downstream analysis Fig. 5e and onwards, including cell:cell interaction data must be considered with caution. I suggest that this section be deleted from the paper. 3. On a related matter, it is simply impossible to conclude given the time points for TAC used, that ECM-Fib form without proliferation or via a myofibroblast intermediate. This conclusion should be substantially qualified.

In other respects this is a useful atlas.

REVIEWERS' COMMENTS

Reviewer #1 (Remarks to the Author):

The authors have responded my qu'rations and the majority of complex issues raised by other reviewers. I believe that the manuscript remains descriptive but it certainly adds important new expressional information to generate new hypotheses.

Thank you!

Reviewer #2 (Remarks to the Author):

The authors have addressed my main concerns. I have no further recommendations.

Thank you!

Reviewer #3 (Remarks to the Author):

Please see my comments below. I am happy to support publication but am rather on the fence about it. Other weaknesses not described below are the obviousness and lack of novelty of many findings, the general lack of new biological insights or hypotheses and general failure to depart from the ingrained model of fibrosis as a general ramping up (and down) of ECM gene expression. Same for the sections on ECs and VSMCs.

Overall, there are some useful comments and findings, strengthened by the use of multiple Cre lines and ability in the end to collapse different lineage subtypes down to a large pool of cells for comparative study. I see it more as an atlas than an insightful paper.

**

The authors have added substantially to the previous draft and in particular have added a later time point for the TAC model. Much of the draft has been rewritten!

We thank the reviewer for the overall positive evaluation of our revised work. We agree that some of the data we present are just computational models and we have now explicitly stated in the manuscript that these have to be interpreted with caution.

Weaknesses still remain in this paper.

1. I don't see why the authors could not have used their VSMC/pericyte Cre drivers to ask to what extent mural cells contribute to ECM-Fib in this TAC model.

We apologize for this. Due to the difficult working conditions related to the pandemic situation where we had to reduce our mouse colony substantially combined with bad breeding performance of the VSMC/pericyte Cre drivers, we were not able to get enough mice for a later time point of the mural cell population. Regarding our presented results of the time point 14 days after TAC, we do not see any contribution of mural cells to ECM-Fib or fibroblast in general, as shown by Myh11CreER and Ng2CreER based fate tracing. Cells from these Cre drivers do not contribute to the fibroblast cluster (Fig. 1d). The missing late time point for the mural cells is part of the discussion.

2. I disagree that the time points chosen represent early phases of pressure overload. EF is already substantially reduced at 2 weeks. There are several implication, among them whether addition of the late time point adds very much. The main finding is the ECM-Fib is increased from day0-14 and day14-28, making it the main focus of attention among fibroblasts. However, I wonder whether the trajectory analysis is meaningful. Fig. 5a is really not compelling if the claim is the trajectories indicate formation of ECM-Fib from Fib1. The same goes for the associated supplementary figure. I do not know how the authors decided to place heavy emphasis on this as opposed to alternative hypotheses (if earlier time points might be included). I cannot comment on how the monocle was set up as in Fig. 5c,d so as to focus on this transition; however, I suggest that this is focus is not justified based on results in Fig.5a. In any case, the downstream analysis Fig. 5e and onwards, including cell:cell interaction data must be considered with caution. I suggest that this section be deleted from the paper.

Since one of the technical strengths of this study is the combination of inducible genetic fate tracing of specific cardiac cells populations, we decided in the initial design of the experiment to generate data at a time point where hypertrophic remodeling already occurred to some extent, while also still being an active remodeling process. As all reviewers pointed out in the first revision, generating data from a second time point would potentially allow us to draw more conclusions on the progression of the cardiac remodeling process. Therefore, we generated data at a later time point for fibroblasts and endothelial cells. It is true that the ejection fraction is already reduced two weeks after TAC surgery, however the hypertrophic tissue remodeling is still progressing which is clearly shown by the continuously elevated heart weight (Fig. 1c). Objectively defining an early and late phase after TAC is rather difficult. Regarding functional decline measured by echocardiography, the "early phase" is probably already over two weeks after TAC. Therefore we carefully rephrased our statements regarding an "early phase" after TAC and in addition added the following statement to the discussion: *"Furthermore, we need to point out that we lack the acute phase directly after TAC surgery."*

Currently the cellular origin of disease associated ECM-Fib is unknown. We can only say with sufficient certainty that a substantial part of this subtype originates from a pool of preexisting fibroblasts. Our RNA velocity analysis aims to get an unbiased insight regarding this origin. We analyzed each condition separately and observed an overall trend of vectors towards Atf3-Fib, Fib2 and Fib3 in each of the conditions. This lead to the assumption that these subtypes might rather reflect end points of differentiation. An origin of vectors could be observed between cluster Fib1, Fib2 and ECM-Fib for sham and this origin seemed to be shifted more towards Fib1 in TAC 14 and TAC 28. Furthermore, latent time seems to indicate high transcriptional turnover in Fib1 increasing to some extent after TAC. Furthermore, Buechler et al. recently reported a fibroblast population marked by Col1a15 as a cross-organ intermediate fibroblast subtype, between a potential Pi16+ progenitor population and different specifically differentiated subtypes (Buechler et al., Nature 2021, PMID: 33981032). Col1a15 was found as the highest enriched marker gene for Fib1 (Fig. 3d), providing further evidence for Fib1 as ECM-Fib precursor. As ECM-Fib continuously increased in number after TAC (Fig. 3f), they potentially represented a more terminal differentiated subtype and were selected as the endpoint of the trajectory. Taken together, we found this evidence as sufficient basis to model a differentiation trajectory from Fib1 to ECM-Fib. We agree with the reviewer that this analysis needs to be validated by further *in vivo* experiments. We have now stated explicitly in the manuscript that this is just a model based on the data.

We added the following parts to the results section:

"Buechler et al. recently reported a fibroblast population marked by Col1a15 as a cross-organ intermediate fibroblast subtype, between a potential Pi16+ progenitor population and different specifically differentiated subtypes. Col1a15 was found as the highest enriched marker gene for Fib1 (Fig. 3d), providing further evidence for Fib1 as ECM-Fib precursor. As ECM-Fib continuously increased in number after TAC (Fig. 3f), they potentially represent a more terminal differentiated subtype and therefore were selected as the endpoint of the trajectory."

"Of note, the differentiation trajectory described above should be interpreted as a model, which is supported by our data."

We added the following part to the discussion:

"Lastly, trajectory analyses and cell-cell communication analyses represent computational predictions, thus interpretation must be considered with caution."

3. On a related matter, it is simply impossible to conclude given the time points for TAC used, that ECM-Fib form without proliferation or via a myofibroblast intermediate. This conclusion should be substantially qualified.

Since there is a significant expansion of ECM-Fib between our two weeks and four weeks time point, we think it is possible to conclude that ECM-Fib increase in number by differentiation rather than proliferation, as no proliferating fibroblasts were found in the sequencing data at two weeks. It seems unlikely that we miss an myofibroblast intermediate between two and four weeks therefore we hypothesize based on our data that ECM-Fib expansion is by differentiation rather than proliferation. However, we agree with the reviewer that we have to be careful with the interpretation and therefore we have added the following sentence to the manuscript:

"However, we cannot exclude that proliferation of pre-existing ECM-Fib occurs outside of the time-points covered by our scRNA-seq analysis."

In other respects this is a useful atlas.

We thank the reviewer for the kind words on the utility of our perivascular map.